

# In-Tandem Multi-Waveband Particulate Absorption and Size Observations Yield Substantial Increase in Radiative Forcing over Industrial Central China

*Luoyao Guan[1], Jason Blake Cohen[1, *], Shuo Wang[1, 2], Pravash Tiwari[1], Zhewen Liu[1], Kai Qin[1]*

[1]Jiangsu Key Laboratory of Coal-Based Greenhouse Gas Control and Utilization, School of Environment and Spatial Informatics, China University of Mining and Technology, Xuzhou, China
[2]Carbon Neutrality Institute, China University of Mining and Technology, Xuzhou, 221116, China

*Corresponding to:* Jason Blake Cohen (jasonbc@alum.mit.edu)

**Abstract.** Coal-based industry in Shanxi, China, including power generation, steel, coke, and chemical manufacturing, emits large quantities of black carbon (BC), contributing significantly to regional aerosol radiative forcing. However, there are substantial scientific uncertainties in the radiative properties of the aerosols in these types of regions due to multiple sources of BC and high emissions of co-emitted aerosol precursors, producing mixed aerosols of different ages, sizes, and morphologies. This study combined optical particle size and multi-band in-situ BC mass and column aerosol optical depth, with MIE modeling to simulate optical properties per particle and over the atmospheric column for absorbing aerosols. These results are applied in a radiative transfer model to constrain regional radiative forcing. First, BC shows a trimodal fine-mode (size<2.5 µm) size distribution, substantially differing from current assumptions of aerosol size made by satellite and atmospheric modeling communities. Second, the coating ratio between absorbing-core and refractive-shell varies dynamically, challenging the widely used fixed mixing ratio assumption. Thirdly, absorbed solar radiation under 500nm is weaker than from 500 to 700nm, and weaker still than above 800nm, challenging assumptions of flat or decreasing absorption with radiative band. Our results yield a reduced single scattering albedo (-0.049 to -0.008) and substantial change in column number (-$1.73\times10^{12}$ to $5.74\times10^{10}$ # m$^{-2}$), resulting in radiative forcing from 0.3 to 3.0 W m$^{-2}$, surpassing local $CO_2$ and $CH_4$ forcing. This work provides a realistic probabilistic framework to quantify BC aging and mixing induced optical properties in industrial regions.

## 1 INTRODUCTION

Aerosols are among the most important forcing agents contributing to anthropogenic global change (Bond et al., 2013; Ramanathan and Carmichael, 2008; Jacobson, 2001), impacting climate through direct (Paulot et al., 2018), semi-direct (Ackerman et al., 2000) and indirect (Lohmann et al., 2000; Koch et al., 2009; Twomey, 1974; Garrett and Zhao, 2006) radiative effects, leading to a substantial influence on earth's radiation and energy balance (Jacobson, 2001). The chemical composition, optical and microphysical properties of aerosols vary widely due to factors such as source type and magnitude, in-situ processing, and interaction with the environment, leading to a broad range of properties with considerable uncertainty in terms of their climate effects (Bellouin et al., 2020; Li et al., 2025). A subset of aerosols strongly absorbs incoming solar radiation (Ramanathan et al., 2001; Chen et al., 2022a), and therefore have a further impact on the atmospheric energy balance. The most important of these absorbing aerosols (AA) from the perspective of the net absorption of solar radiation across the spectrum on a per particle basis is black carbon aerosol (BC) (Wang et al., 2021a; Tiwari et al., 2023; Koch et al., 2009).

BC is emitted over industrial regions together with multiple co-emitted semi-volatile species due to the incomplete combustion of carbon-based fossil fuels and large amounts of heat. There are vastly different emissions profiles from direct incomplete combustion associated with energy production (Mousavi et al., 2019) and heat plants (Pirjola et al., 2017), compared with combustion for smelting, redox reaction and other materials including steel and other metals (Zong et al., 2016), chemical plants (Cao et al., 2006), coking (Mu et al., 2021), and related industries (Mousavi et al., 2018). Furthermore, such regions



also have large amounts of coal dust aerosol- composed predominantly of carbon which is optically similar to BC (Khan et al., 2017) generated during the mining, transport and storage of coal (Csavina et al., 2014). Key microphysical properties including size and mixing state of these emitted BC aerosols varies significantly with the type and amount of coal used, combustion technologies, in-situ processing after emissions and before observation, and other environmental factors (Bond et al., 2013). Ground-based studies often prioritize matching aerosol optical depth (AOD) observations (Holben et al., 1998; Wei et al.,

2024) or use the measured mass concentration (Savadkoohi et al., 2024; Saarikoski et al., 2021) or single particle properties after altering their shape through heating or chemistry (Wang et al., 2021b; Romshoo et al., 2022) to derive optical properties like single-scattering albedo (SSA) and asymmetry parameter (ASY). Remote sensing studies and models adopt a single lognormal fine mode distribution to improve statistical performance of size over less complex environments (Omar et al., 2005; Reddington et al., 2013; Shen et al., 2019), leading to biases compared with more detailed observed distributions (Junker et

al., 2006; Wu et al., 2020). While in-situ measurements provide more precise size distributions, they typically rely on a single waveband to approximate optical properties (Wang et al., 2023a; Huang et al., 2024). Those studies that employ multi-band observations currently apply a fixed size distribution (Prats et al., 2011; Yu et al., 2012). Due to high levels of co-emitted oxidants (e.g., $SO_2$, $NO_x$, $NH_3$) and water vapor, atmospheric aging of BC and other primary AA occurs rapidly in industrial basins (Guo et al., 2024; Qin et al., 2023; Li et al., 2023a; Lu et al., 2024). Therefore the current community emphasis on

overly simplified BC mixing states including approximation using an external mixture (Lesins et al., 2002; Moosmüller et al., 2011) or fractal approximation (Yuan et al., 2019; Loh et al., 2012), neglect the aging processes (coagulation, condensation) occurring in the real world (Guan et al., 2024), which lead to more thermodynamically stable core-shell structures (Lu and Bowman, 2010).

Existing tools to account for total column properties of aerosols rely on iterative tools like the Optical Properties of Aerosols

and Clouds (OPAC) framework (Srivastava et al., 2012; Bibi et al., 2017). However, OPAC's rigid parameterization lacks flexibility to represent realistic variations in particle size and mixing states, as well as adversely producing non-unique outputs—critical factors influencing and adding uncertainty to the aerosol radiative effects associated with BC in general. Other approaches assume a fixed core size or property and then use this to assimilate ground and satellite observations of total extinction (i.e., AOD), which is based on a simplification of BC's optical properties assuming a fixed Mass Absorption Cross-

section (MAC) of ~7.5 $m^2$/g at 550 nm (Randles et al., 2017). Even more recent studies which use a somewhat varying MAC (Brown et al., 2021) neglect the range of observed enhanced absorption and simultaneous changes in scattering from internal mixing and lensing, which can theoretically impact absorption by 50-100% (Bond and Bergstrom, 2006; Lack et al., 2009). Absorbing aerosols significantly impact regional climate by absorbing solar radiation and increasing scattering, causing heating in the lower and middle atmosphere, altering environmental and energy cycles (Barbaro et al., 2013; Chung and Zhang,

2004; Fan et al., 2012). Since BC is a primary aerosol, it tends to be distributed throughout the lower and middle troposphere (Bond et al., 2013; Liu et al., 2015; Cohen et al., 2018; Wang et al., 2020b), intensifying atmospheric warming (Tian et al., 2020). The quantification of radiative forcing estimates is a widely used approach to understand this BC-radiation interaction and impact (Myhre et al., 2014; Ramanathan and Carmichael, 2008; Bond et al., 2013). However, existing model-based studies contain significant uncertainties, with studies reporting a broad range of radiative effects, from slightly cooling -0.1 to

substantial warming 2.0 W $m^{-2}$ (Tiwari et al., 2023; Cohen and Wang, 2014; Chen et al., 2022b; Chung and Seinfeld, 2005). One of the major issues surrounding these uncertainties is that these calculations depend on the specific physical and optical characteristics of BC in-situ, with it being essential to elucidate both per-particle properties and column number loadings in tandem, in order to match observed variability both at the surface and through the column of aerosol absorption (Liu et al., 2024b; Bao et al., 2020; Kahn et al., 2023). Recent studies have demonstrated the persistent uncertainties in BC radiative

forcing estimates, emphasizing challenges in addressing per-particle optical properties and underutilized synergies across multi-waveband remote sensing data (Peng et al., 2016; Matsui et al., 2018). A recent study has explored the discrepancy induced by this rigid parameterization in morphology (Tiwari et al., 2023) demonstrating that over 1.9% of solutions using



internal mixtures of BC and sulfate in tandem and flexible sizes of BC and sulfate shell, when constrained by multi-waveband SSA observations and uncertainties, could yield a net positive TOA (at the top of the atmosphere) forcing, highlighting sensitivity to morphology and multi-waveband constraints. Similarly, reanalysis platforms like MERRA-2 and CAMS face similar challenges in accurately quantifying BC radiative forcing due to oversimplified optical, physical parameterizations and column loadings (Fu et al., 2022).

To address the aforementioned challenges, this work employs measurements of particle number and multi-band in-situ measurements of BC mass and column AOD, over a coal-based industrial region in Shanxi Province, China, which is typical of many such industrial areas around the world and especially so in the Global South. Using MIE scattering, we quantify how dynamic variability of aerosol size distribution and mixing state affect per-particle optical properties. Then we use these derived microphysical solutions to invert corresponding BC column number, mixing, and mass concentrations. Finally, the derived suite of observationally constrained probabilistically determined solution of size and mixing state and corresponding optical properties are used to drive a radiative transfer model to estimate local radiative forcing contributions. This integrated approach establishes a physically grounded framework for quantifying BC-climate direct effect in complex emission environments and informs improved parameterization for AA processes in terms of both their regional and global climate impact.

## 2 Data and Methods

This work is based on an extensive suite of observations made in an intensive coal-based production and industrial use region in Shanxi province (36.07° N, 112.88° E) in August 2022 (Fig. 1). This location was selected for its proximity to a rapidly evolving industrial landscape characterized by intensive coal mining (more than 200 coal mines) and a wide range of coal-consuming industries, including power generation, coking, steel, and chemical manufacturing, alongside traditional agricultural activity and various small industries such as brick making. Additionally, the basin-like topography of the region promotes the accumulation and prolonged atmospheric residence of BC, facilitating complex interactions between primary aerosols and secondary aerosol precursors and hence aging of BC aerosols. This combination of multiple sources within a confined spatial domain makes the region highly representative of many rapidly developing industrial areas across the Global South (Tiwari et al., 2025; Ramachandran et al., 2023).

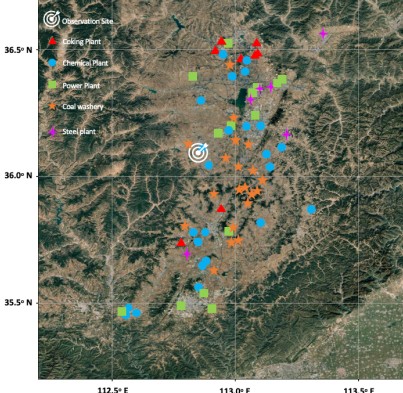

**Figure 1: The map of the geographical region of industrial covered in this study (Base map data © Google Earth, Maxar Technologies).**



### 2.1 Ground-based Observations

This study employs a suite of ground-based instruments to quantify the mass concentration, size distribution, and optical properties of aerosols (Fig. S1). These synchronized, multi-parameter observations form the basis for characterizing black carbon (BC) and AA properties across both physical and optical domains.

A continuous BC mass concentration detector (Aethalometer AE-31) is utilized to compute BC mass information based on the strong absorption characteristics of BC across various wavebands(Yang et al., 2009), spanning from ultraviolet (UV) to near-infrared (NIR): 370, 470, 520, 590, 660, 880, and 950 nm. The instrument operates using optical attenuation and time-differential techniques, providing 5-minute resolution data in ng m$^{-3}$. In conjunction with the Aethalometer, the study also incorporates the GRIMM-180, which continuously measures particle size distribution using a light scattering technique(Liu et al., 2018). This instrument provides size information (# L$^{-1}$) across 32 size bins, ranging from 0.25 μm to 32 μm in diameter. For this work, we focus on the number concentration measurements in the fine mode (Dubovik et al., 2000) over 14 size bins from 0.25 μm to 2.0 μm in diameter, while maintaining a temporal resolution of 5 minutes. Additionally, a portable solar photometer (Microtops II) is employed to measure in-situ aerosol optical properties and atmospheric conditions based on the Beer-Lambert law (Ichoku et al., 2002) including AOD and water vapor content. This device operates at five specific wavebands (440 nm, 500 nm, 675 nm, 870 nm, and 936 nm) and is used to capture the temporal variation of AOD at different wavelengths. By integrating data from these three instruments, this study is able to utilize information of mass concentration, size distribution and aerosol optical properties in tandem.

To ensure data reliability, a strict quality control protocol was applied. First, all physically unrealistic values were removed. Second, data points were removed if they were two or more standard deviations away from the set of five data points including the value in question as well as the two preceding and following measurements, ensuring temporal continuity and consistency in the dataset (Wang et al., 2023b). Following filtering, a total of 1410 quality-controlled, time-synchronized measurements were obtained for BC mass concentration at all five wavelengths as well as particle size distribution diameter range from 0.25-2.0 μm. These datasets were then merged with direct sun observations of AOD, so as to provide a consistent set of measurements to robustly analyze and assess aerosol optical, mixing, size, and number properties in tandem, and their impacts on aerosol loading and radiative forcing, in a complex environment.

### 2.2 Parameterization of AA Mixing State

To represent the mixing state of AA in the industrial region, this work adopted two distinct and opposite assumptions to allocate observed BC mass, which comprehensively and robustly span the physical possibilities of realistic mixing state within the atmosphere in order to (Kim et al., 2008). The uniform assumption is when each particle size bin has an equal mass of BC, and physically represents the behavior of AA freshly emitted into any given atmospheric parcel from a variety of sources. This is used as a major assumption by most of the current generation of chemical transport models, including but not limited to (Croft et al., 2024; Gaydos et al., 2007), as well as most optical aerosol models currently used by the satellite community (Wang and Martin, 2007) as well as the OPAC modeling system (Fillmore et al., 2022). In contrast, the non-uniform assumption means the mass ratio of BC-to-particulate mass across all size bins is the same. This assumption implies that aerosols of all 14 size bins are composed of the same components, and are in a current state of complete internal mixing, which is relevant for aerosols which have been in the atmosphere long enough to undergo condensation- and coagulation-based growth and atmospheric in-situ chemical, physical, and water-based processing. This is the assumption currently adopted by most sun-photometer networks AERONET (Kayetha et al., 2021; Schuster et al., 2005) and SONET (Wang et al., 2013) as well as a small number of research-based chemical transport models (Kim et al., 2008; Wang et al., 2023a).

Due to the fact that the fineness of the sizes observed is at a higher resolution compared with existing regional and global transport and climate models (Vignati et al., 2004; Randles et al., 2013; Kokkola et al., 2018), we introduce a commonly used set of assumptions to reduce the data of the size into a set of analytical functions that still allow for comprehensive analysis



which is comparable with existing assumptions and standards, following (Cohen et al., 2011; Wang et al., 2023a). In specific, this study employed three different particle size distributions: (1) ISSIZE, which used the observed in-situ size distribution data; and the two analytical approaches, which were fitted based on the observed size distributions (2) $Log_1$, which represents the best-fitting single-peak lognormal (Seinfeld et al., 2003); and (3) $Log_{123}$, which represents the best-fitting set of three single-peaked lognormal distributions summed in tandem. By integrating these two mixing assumptions and three size distribution schemes, this study establishes a flexible and observation-constrained framework to evaluate how assumptions about particle morphology influence derived aerosol optical properties including the extinction, scatter, asymmetry, and single scatter albedo (SSA) per particle as well as resulting net top of atmosphere radiative forcing (TOA).

### 2.3 MIE Model

MIE scattering theory provides an exact solution for the interaction of a plane electromagnetic wave with a uniformly sized spherical particle in a homogeneous medium, enabling precise quantification of light absorption and scattering(Wang et al., 2021a). The MIE model has been extensively validated for key atmospheric aerosols, such as sulphate and black carbon, which often assume spherical or nearly spherical shapes after equilibrating with the in-situ environment(Liu et al., 2020). In addition, when very fresh such particles are observed to be non-spherical, they can still be estimated by a spherical MIE model, just yielding a slightly different set of core and shell size parameters (Mishchenko et al., 1997). Its applicability for particles whose sizes are comparable to the wavelength of light ensures the reliability of computing parameters like SSA, as observed in previous remote sensing research (Dubovik and King, 2000).

In this study, the MIE model was employed based on the core-shell assumption, where strong absorbing substances such as BC and coal dust (this area only includes these two absorbing sources in any significant amount) due to their higher density, were assumed to be the core, and non-absorbing substances such as sulphate, nitrate, ammonium, and aqueous aerosol water were assumed to be the shell. This configuration reflects the dominant emission composition from surrounding coal-producing and coal-consuming industries, including $SO_2$, $NO_x$, water vapor, and methane (Li et al., 2023b; Hu et al., 2024).

The size information of core and shell was obtained by calculating the proportion of specific BC mass observed at each wavelength i (470 nm, 520 nm, 660 nm, 880 nm, and 950 nm) in each size bin. The observations of AOD at each wavelength j (440 nm, 500 nm, 675 nm, 870 nm, and 936 nm) were used as inputs in the model. The refractive indices of the core ($m_1$=2+1i) and the shell ($m_2$=1.52-5×10$^{-4}$i) were applied in the MIE model to compute key optical parameters: extinction ($EXT_{ij}$), absorption ($ABS_{ij}$), scattering ($SCA_{ij}$), and asymmetry parameter ($ASY_{ij}$) on a per particle basis, considering the size, observed wavelength of the BC (i) and observed wavelength of the AOD (j), providing a full representation of the range of possible values of AA in the area observed.

To ensure consistency, we only analyze combinations of i and j in which the wavelength of the AOD and BC mass concentrations are closest, ensuring consistency in the calculations. For example, the AOD observed at j=440 nm is paired with BC mass observed at i=470 nm, hereafter labeled as AOD440BC470 or λ=470. This combination of values allows consistency across all of the different observational platforms.

### 2.4 Calculation of aerosol column number loading

The particle column loading (# m$^{-2}$) is computed following Eq. (1) and Eq. (2) following (Cohen and Wang, 2014):

$$\varepsilon_w = \frac{\sum_{k=1}^{14} \varepsilon_{k,\lambda} n_k}{\sum_{k=1}^{14} n_k} \tag{1}$$

$$N_\lambda = \frac{AOD_j}{\varepsilon_w} \tag{2}$$

where $\varepsilon_{k,\lambda}$ is the extinction coefficient (μm$^{-2}$) per particle in each size bin k, based on the value computed at the matched waveband λ (as defined in Sect. 2.3) $n_k$ is the concentration (# L$^{-1}$) of particles in each size bin k, $\varepsilon_w$ is the weighted extinction



coefficient per unit volume, AOD is the observed aerosol extinction per band j (as defined in Sect. 2.3), and $N_\lambda$ is the column number loading of particles per matched band $\lambda$.

### 2.5 Santa Barbara DISORT Atmosphere Radiative Transfer (SBDART)

SBDART is employed to simulate atmospheric solar irradiance under two scenarios: with and without the presence of aerosols (Ricchiazzi et al., 1998; Tiwari et al., 2023). Given that the majority of incoming solar radiation falls within the spectral range of 0.25 to 4.0 μm, this work follows the general community and focuses its analysis over this specific region of the electromagnetic spectrum (Ma et al., 2021).

To perform irradiance calculations, SBDART requires several key input parameters. These include column AOD (as observed),
per particle SSA and ASY (as derived from the MIE model), and additional data of precipitable water, surface spectral albedo, and column ozone data, all of which are sourced from ERA-5 reanalysis (Hersbach et al., 2020). Meteorological data, such as temperature, pressure, and ozone profiles at various pressure levels required to setup the model atmosphere is also sourced and prepared from the ERA-5 reanalysis.

The calculations for aerosol direct radiative forcing (DRF) using SBDART model are well-established in the literature
(Ricchiazzi et al., 1998). Specifically, the upward and downward irradiances are determined at both the top of the atmosphere and the surface. From these values, the differences between the irradiances with and without aerosol effects on a diurnal basis are retained as TOA (at the top of the atmosphere) and BOA (at the surface). Finally, the net atmospheric forcing is derived as the difference between TOA and BOA. Detailed formulations of these calculations can be found in previous studies (Tiwari et al., 2023; Wang et al., 2020c).

The net TOA is simulated separately for the two mixing assumptions, based on the various amounts and properties of BC estimated across the different spectral observations. By analyzing the radiative forcing across all of these scenarios, the impacts of how different aerosol mixing and different optical observational states impact the overall energy balance of the atmosphere is explored.

### 2.6 Statistics and Fitting

The efficacy of the for rapid radiative forcing correction least squares modeling fits are computed based on a combination of three statistics: $R^2$ ($p < 0.05$), which represents the point-by-point reproducibility; RMSE (W m$^{-2}$), which is a measure of the radiative forcing uncertainty between the observations and the model; and the range of values reproduced (ratio of the range of calculated average observed radiative forcing to the range of each individual size bin observed radiative forcing), which indicates the precision of the total range of the values reproduced. Statistical comparisons of differences between observed
datasets are quantified using a two-tailed student-t test with two different statistical levels of certainty ($p < 0.05$) and ($p < 0.10$).

## 3 RESULTS

### 3.1 Analysis of AA Physical Properties

More than 95% of size observations contain a three-peaked size distribution, located around 0.25 μm - 0.28 μm, 0.58 μm - 0.65 μm, and 0.7 μm - 0.8 μm respectively, as shown in Fig.2. To analyze the variability in the three-peaked size distribution,
we categorized the ratio between the first and second peaks using statistical percentiles. The maximum ratio represents extreme cases where the first peak dominates, whereas the 25th, 50th, and 75th percentiles capture the progressive shift in dominance between the first and second peaks. This approach allows for a more comprehensive representation of different tri-modal distribution patterns observed in the dataset. This approach also forms a basis by which it can clearly be observed that there are different contributing factors to the BC emissions in the area including but not limited to high temperature combustion for





power and steel factories, low temperature combustion for chemical factories and local heating systems, and coal dust due to the thousands of coal trucks daily plying the roads(Li et al., 2023b; Qin et al., 2023).

To provide consistency with past methods, a set of three lognormal fits of the distribution are made. The first mode (FM) captures details from 0.25 µm to 0.5 µm, the second mode (SM) captures details from 0.5 µm to 0.7 µm, whereas the third mode (TM) captures details from 0.7 µm to 1.6 µm. There are multiple sources of emissions associated with coal mining,

production, transportation, and utilization processes, with each of these sources more strongly related to different observed modes (Fig. S2). Aerosols in FM are likely due to high efficiency combustion associated with power and steel, aerosols in the SM are more likely due to lower efficiency combustion associated with boilers, chemicals, and coking, whereas aerosols in the TM are more likely due to coal production and dust. As observed herein, a single sub-PM$_{2.5}$ size distribution with simple source type is never adequate to capture the observed size distribution pattern.

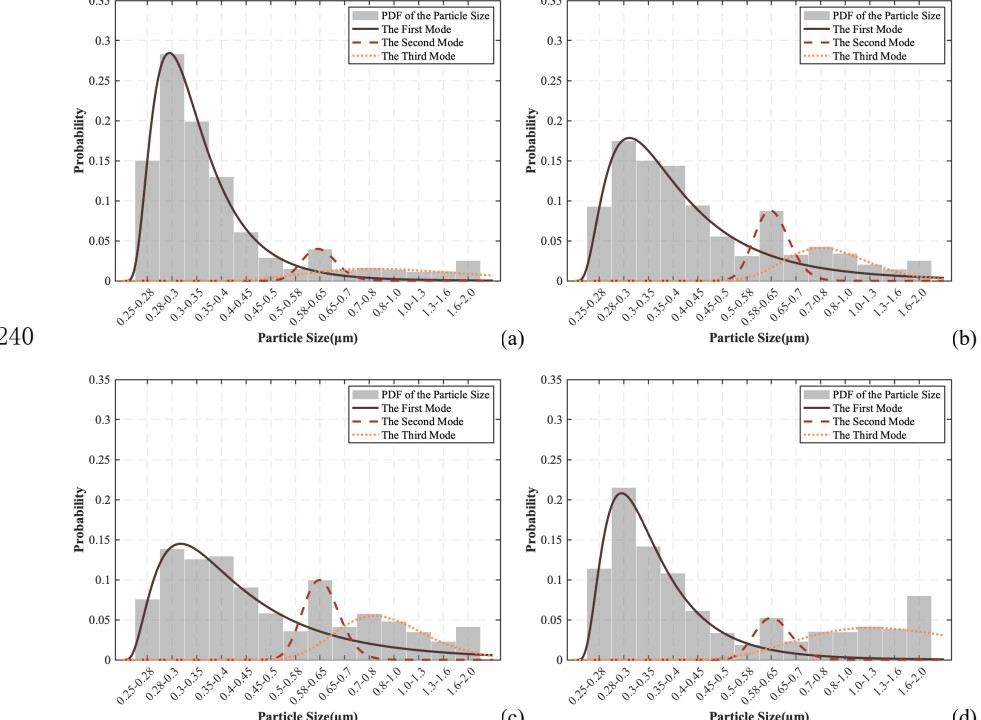


**Figure 2: Variability in the size distribution across all 14 bins at different times, categorized based on the ratio between the first and second peaks in the tri-modal distribution. (a) maximum ratio between the first and second peak; (b) 25th percentile ratio between the first and second peak; (c) 50th percentile ratio between the first and second peak; (d) 75th**
**percentile ratio between the first and second peak.**

### 3.2 AA Optical Properties

The MIE model is used to compute the observationally constrained per-particle SSA and its dependence on wavelength, size distribution, and mixing state (Fig. 3), illustrating these effects through vertical comparisons of different size distributions and horizontal comparisons of mixing state assumptions. While it may be possible to use a common absorbing angstrom exponent
(AAE) to fit SSA over specific wavebands and subsets of size (Helin et al., 2021), the results break down across both the entire size range of interest, as well as across wavebands even in the same size range solution set, as described in Fig. S3. This finding lends further credence to the results outlined in this work, since the majority of present models and observations make the assumption that the AAE is a valid approximation across different wavebands (Wang et al., 2020a).




Longer wavelengths (880 nm, 950 nm) show a stronger per-particle absorption than shorter wavelengths (470 nm, 520 nm),
with observations at 660 nm usually in the middle for smaller sized particles. Although the 1-standard-deviation ranges of SSA
start to overlap for 0.65 μm and larger particles, there is still a consistent per-particle positive SSA bias computed at higher
wavelengths when comparing the results for all combinations of mixing state and size against the observed ISSIZE and uniform
mixing results.

In term of the size distribution, per-particle SSA is consistently and slightly lower for the $Log_{123}$ approximation as compared
to ISSIZE, and lower still for the $Log_1$ approximation (Fig. 3). ISSIZE consistently shows a one-standard-deviation over the
mean per-particle absorption which is the least across all assumptions, especially at 470 nm.

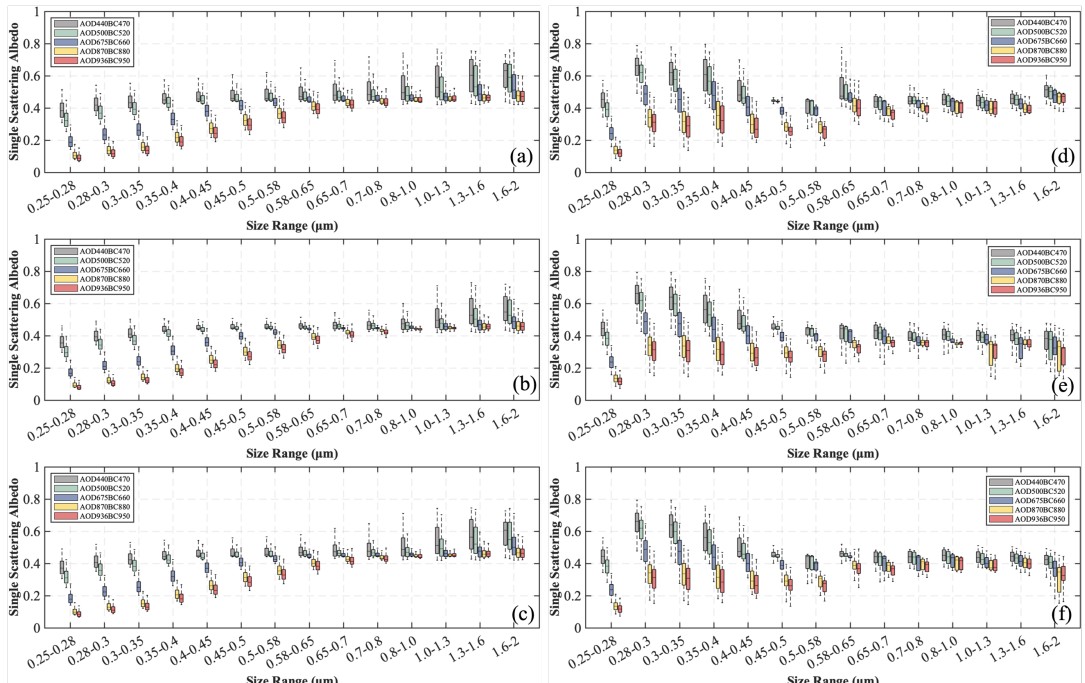

**Figure 3: Multi-wavelength SSA simulation results under different assumptions and particle size distributions. (a) non-uniform assumption with ISSIZE distribution. (b) non-uniform assumption with $Log_1$; (c) non-uniform assumption with $Log_{123}$; (d) uniform assumption with ISSIZE distribution; (e) uniform assumption with $Log_1$ (f) uniform assumption with $Log_{123}$.**

The difference of simulated multiband SSA under different assumptions across all sizes are shown in Fig. S4, demonstrating
that all simplifying assumptions of particle size or single waveband yield stronger per-particle absorption compared to the
observations. Furthermore, the results also indicate that this underestimation becomes more pronounced at larger particle sizes.
Therefore, applying $Log_{123}$ yields a less biased approximation of the actual size distribution's results compared with applying
the $Log_1$ assumption, especially when considering aerosol mass. Specifically, when considering the non-uniform assumption,
at 440 nm, $Log_1$ and $Log_{123}$ have bias compared to ISSIZE in FM of 0.20 and 0.10, respectively, which increase in SM and
TM to 0.29 and 0.14, respectively. This suggests that the underestimation of SSA is more significant for larger particle sizes
(still within the fine mode). Therefore, it is essential to effectively model particles with diameters ranging from 0.65μm -
1.30μm to reduce the per-particle absorption bias, and supports that a trimodal simplifying assumption is more realistic than
the single mode.

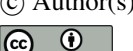



### 3.3 Column Properties and TOA Radiative Forcing of AA

Seven days of AOD observations at 440 nm, 500 nm, 675 nm, 870 nm, and 936 nm are made using Microtops II. The

differences in AA assumptions lead to non-negligible differences from $4.8 \times 10^{12}$ m$^{-2}$ to $3.3 \times 10^{13}$ m$^{-2}$ and from $1.1 \times 10^{8}$ ng m$^{-2}$ to $9.2 \times 10^{8}$ ng m$^{-2}$ of in-situ number and mass column loading. These changes are larger than the range of simulation ability that current chemical transport models have derived as demonstrated by reanalysis products(Tiwari et al., 2025; Liu et al., 2024b; Liu et al., 2024a) and model studies using common emissions inventories and standard aerosol processing and radiation packages(Gaydos et al., 2007; Croft et al., 2024). This range is consistent with one or more issues still to be addressed by the

aerosol modeling community: insufficient knowledge or assumptions about in-situ aerosol size distribution, unclear mechanisms of in-situ processing including but not limited to aerosol mixing and long-range transport, and inaccurate estimation of emission magnitude and location.

The radiative forcing of AA with different mixing states and assumption at TOA are calculated based on the weighted number concentration for each size bin, as shown in Fig. 4. The range of radiative forcing with internal mixing varies from 18.3 - 29.9

W m$^{-2}$ (ISSIZE), 18.5 - 30.5 W m$^{-2}$ (Log$_{123}$), and 19.0 - 31.1 W m$^{-2}$ (Log$_1$) as compared to the respective ranges of externally mixed aerosol which range from 3.6 - 13.4 W m$^{-2}$ (ISSIZE), 3.5 - 13.3 W m$^{-2}$ (Log$_{123}$), and 3.4 - 13.1 W m$^{-2}$ (Log$_1$) when computed at 880 nm. This aligns with the known fact that the lensing effect increases the absorption of light by aerosols on a per-particle basis, leading to stronger positive radiative forcing, and that the total column number of AA is also a necessary and extremely important observational result required to provide a reasonable calculation of radiative forcing in the areas

observed herein. Since internally mixed aerosols are more commonly observed in-situ where our observations are made, due to high pollution levels and low rainfall, allowing secondary production and aging in-situ to occur, all computations of radiative forcing hereafter only consider this mixing assumption.

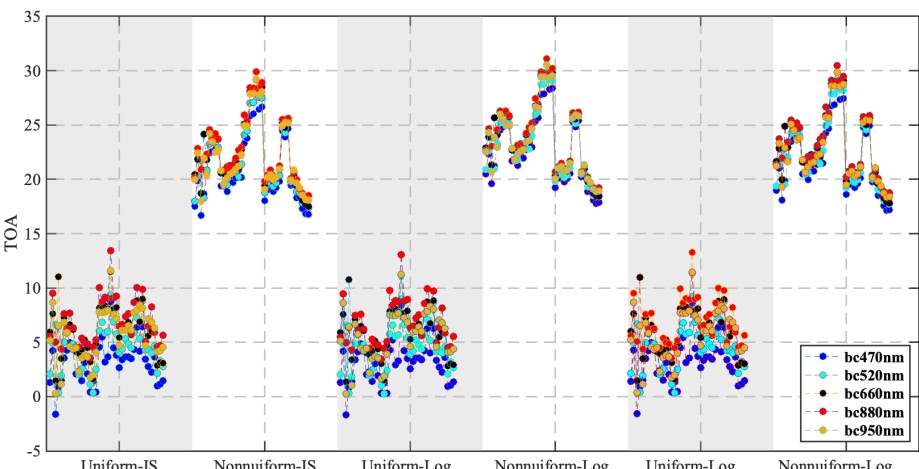

**Figure 4: Temporal variability of radiative forcing under different assumptions from Santa Barbara radiative transfer**

**(SBDART). The first column is ISSIZE distribution with uniform assumption. The second column is INSIZE distribution with non-uniform assumption. The third column is Log$_1$ distribution with uniform assumption. The fourth column is Log$_1$ distribution with non-uniform assumption. The fifth column is Log$_{123}$ distribution with uniform assumption. The sixth column is Log$_{123}$ distribution with non-uniform assumption.**

The variations in radiative forcing across different wavelengths exhibit noticeably different values between $\lambda = 470$ and $\lambda =$

$520$, and statistically significant differences between $\lambda = 470$ and $\lambda = 660$ (p<0.05), between $\lambda = 470$ and $\lambda = 880$ (p<0.05) and between $\lambda = 470$ and $\lambda = 950$ (p<0.10). First, near-infrared wavelength derived products generally overestimate TOA forcing, especially for 880 nm, whereas the near-ultraviolet wavelength always yields the lowest value. However, there are still some interesting exceptions where TOA forcing shows a higher value at near-ultraviolet wavelengths, in particular, when the raw data has a higher number concentration of BC overall, and a higher number fraction at small sizes.



This finding is consistent with the recent idea that uses UV spectroscopy from OMI and TROPOMI in connection with visible spectroscopy to improve aerosol in-situ loading(Liu et al., 2024b; Liu et al., 2024a).

The difference in TOA radiative forcing based on the particle size assumption across different wavelengths from 470 to 950 nm are larger in the case of ISSIZE - $Log_1$ (-1.70 W m$^{-2}$, -1.58 W m$^{-2}$, -1.45 W m$^{-2}$, -1.38 W m$^{-2}$, and -1.49 W m$^{-2}$) respectively, than in the case of ISSIZE-$Log_{123}$ (-0.78 W m$^{-2}$, -0.73 W m$^{-2}$, -0.67 W m$^{-2}$, -0.64 W m$^{-2}$, and -0.69 W m$^{-2}$) respectively (Fig.

5). These discrepancies are robust across different wavelengths and all times of day, indicating a consistent bias at all times within this wavelength range. Other results are shown in the supplementary materials (Fig. S5) for completeness.

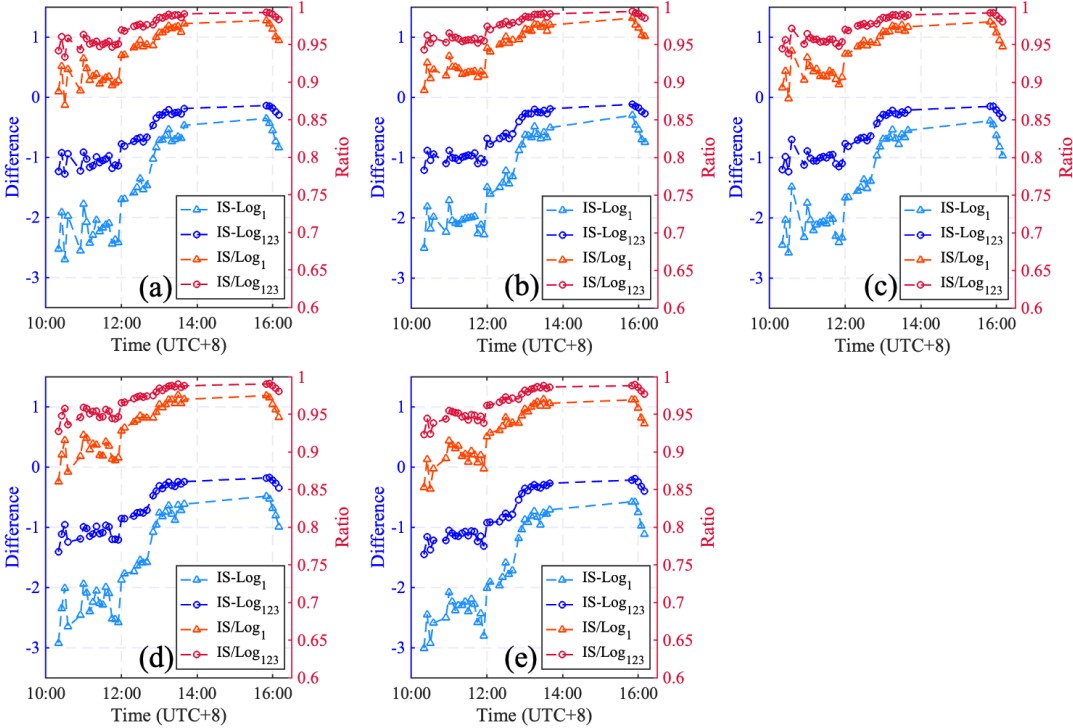

**Figure 5: Differences of multiband radiative forcing and ratio with non-uniform assumption between different size distributions. (a) 470 nm; (b)520 nm; (c)660 nm; (d)880 nm; (e)950 nm.**

Although the bias is smaller in the case trimodal approximation (from 0.2 - 1.5 W m$^{-2}$) than in the single modal assumption (from 0.3 - 3.0 W m$^{-2}$). In this mixing case assumption, the trimodal distribution always aligns better with the true results, as evidenced by the RMSE results in Table 1. It is important to note that there still is some bias included in all cases, however the improvement in AA radiative forcing uncertainty obtained by improving size representation can presents a substantial impact on the net radiative forcing over the area of interest.

**Table 1: The statistic of radiative forcing through all five bands with non-uniform assumption, the mean values of RF are given in bold, the standard deviations are given in parentheses, the last two rows are RMSE between different size distribution.**

|  | 470 nm | 520 nm | 660 nm | 880 nm | 950 nm |
|---|---|---|---|---|---|
| ISSIZE | **20.9**(*3.0*) | **21.6**(*3.1*) | **22.3**(*3.1*) | **22.7**(*3.1*) | **22.1**(*3.1*) |
| $Log_1$ | **22.6**(*3.2*) | **23.2**(*3.2*) | **23.8**(*3.3*) | **24.1**(*3.3*) | **23.6**(*3.2*) |
| $Log_{123}$ | **21.7**(*3.1*) | **22.3**(*3.1*) | **23.0**(*3.2*) | **23.3**(*3.2*) | **22.8**(*3.1*) |
| RMSE(ISSIZE-$Log_1$) | 1.86 | 1.75 | 1.60 | 1.54 | 1.67 |
| RMSE(ISSIZE-$Log_{123}$) | 0.88 | 0.82 | 0.76 | 0.73 | 0.79 |



**3.4 Rapid Radiative Forcing Correction**

To extend the impact of this local work to other coal mining and use sites around the world, a set of first-order linear and non-linear models of radiative forcing are built from our observations of SSA, AOD, and particle size. The point of this approach is to use the breadth of observed values across their entire range of observation to create a fast approximation grounded in the underling physics, optics, and sized measured herein, are observable in other parts of the world using similar or other platforms, and the non-linear processing of SBDART computed radiative forcing, in a way which is more realistic than the current

approaches which rely upon fewer observed values(Mehrotra et al., 2024). While the observed values may not sufficiently explain all such mining and use areas, they are far more representative than no approximation, and may also find use in other more polluted or energy intensive areas in the rapidly developing Global South(Ramachandran et al., 2023). The resulting approximations can be used in global models with high efficiency, as existing models' use of external mixing can be adjusted following the approach herein to account for the improved representativeness of mixing and size on the radiative forcing (Table

S2, Fig. S6).

**Table 2: The statistic of two first-order linear/non-linear models with non-uniform assumption through all five wavelengths, the coefficient of determination is given in bold, the RMSE, and weighted RMSE are given in parentheses (W m$^{-2}$).**

|  | 470 nm | 520 nm | 660 nm | 880 nm | 950 nm |
|---|---|---|---|---|---|
| **ISSIZE** (SSA) | **0.38**(*2.27, 0.20*) | **0.31**(*2.09, 0.21*) | **0.21**(*1.81, 0.22*) | **0.17**(*1.64, 0.24*) | **0.27**(*2.04*), 0.21 |
| **Log$_1$** (SSA) | **0.14**(*1.42, 0.24*) | **0.08**(*1.11, 0.27*) | **0.03**(*0.74, 0.29*) | **0.02**(*0.53, 0.37*) | **0.06**(*1.03, 0.26*) |
| **Log$_{123}$** (SSA) | **0.27**(*3.76, 0.21*) | **0.19**(*3.86, 0.23*) | **0.11**(*4.01, 0.24*) | **0.08**(*4.03, 0.29*) | **0.16**(*3.99, 0.23*) |
| **ISSIZE** (SSA, AOD, size) | **0.83**(*1.77, 0.09*) | **0.79**(*1.84, 0.09*) | **0.75**(*1.90, 0.09*) | **0.74**(*1.91, 0.10*) | **0.77**(*1.94, 0.09*) |
| **Log$_1$** (SSA, AOD, size) | **0.74**(*1.84, 0.10*) | **0.71**(*1.88, 0.11*) | **0.70**(*1.92, 0.12*) | **0.71**(*1.91, 0.12*) | **0.71**(*1.93, 0.11*) |
| **Log$_{123}$** (SSA, AOD, size) | **0.79**(*2.03, 0.09*) | **0.75**(*2.14, 0.09*) | **0.72**(*2.25, 0.11*) | **0.71**(*2.26, 0.11*) | **0.73**(*2.27, 0.10*) |

In all cases (Fig. 6), a more precise fit is obtained when considering the non-linear relationship driven by SSA, AOD, and particle size in tandem. In specific, we find that there is a reduction in the RMSE and simultaneous increase in the $R^2$ at 470 nm, from 2.27 W m$^{-2}$ to 1.77 W m$^{-2}$ and from 0.38 to 0.83 respectively (Table 2). Furthermore, using ISSIZE leads to a further improvement in the fitting, bringing the range of the RMSE of the rapid forcing correction from 1.77 W m$^{-2}$ to 1.94 W m$^{-2}$ (Table 2), which is lower than that of Log$_1$ (1.84-1.93 W m$^{-2}$) and significantly better than Log$_{123}$ (2.03-2.27 W m$^{-2}$). Both

cases are always within 10% of the full model results in Fig. 3.








**Figure 6: Deriving adjusted TOA forcing using two different linear models with non-uniform assumption, the first row in each subfigure is linear model only including SSA, the second row is linear model including the effects of BC core,**





**sulphate shell and AOD as additional variables, the first to the third column for each subfigure is 470 nm, 660 nm and 880 nm. (a) ISSIZE measurements; (b)Log$_1$ distribution; (c)Log$_{123}$ distribution.**

**Conclusions**

The results herein are a first attempt using a suite of surface observations in tandem to analyze the impacts that the multi-wavelength, multi-size, and multi-mixing assumptions have on physically realistic combinations of SSA, particle number,

particle size, particle mixing, and ultimately radiative forcing in a typical industrial area having large amounts of emissions of BC and secondary aerosol precursors from multiple sources. This work ensures that per-particle absorbance is consistent across multiple observed wavelengths in tandem, and that the nonlinear interactions due to dynamic size and mixing state at each waveband are accounted for. Specific findings include reduction in per-particle optical property bias and improved consistency with observations across different wavebands. This active consideration of aerosol size, mixing state, and multiple wavelengths

in tandem yields per particle SSA that is generally higher than represented by traditional modelling and observational approaches, yet is generally lower than obtained by typical satellite-based inversions and standard external mixing optical approaches including OPAC. The results reinforce the concept that modeling studies which do not comprehensively consider these factors will have a per-particle SSA which is too low and hence BC column loading which is too low to match observed column aerosol absorption or (aerosol absorption optical depth) AAOD (Samset et al., 2018; Lee et al., 2016; Kahn et al.,

2023), and conversely that remote sensing-based studies will have an SSA which is too high and therefore will underrepresent the radiative forcing from such regions around the world (Fu et al., 2025).

This work quantifies the per-particle SSA more precisely than approaches using the common community assumption that observations made at a single wavelength with a uniform size distribution and fixed mixing assumption. Additionally, this work challenges the common assumption that applying a mass absorption coefficient (even a varying one) is sufficient, by

clearly demonstrating that such approaches lead to a fundamental underestimation of per-particle SSA in particular for larger particles, which in turn dominate the impact on mass, therefore leading to too low of a BC number concentration to match observed AAOD. The results demonstrate that the total column number loading of AA is the major observational unit that the community must strive to match, in order to successfully compute the radiative effects of AA, in contradiction to studies which change per-particle SSA but do not simultaneously change BC emissions in tandem (Huang et al., 2024).

Another aspect of the results address the well-known issue of radiative forcing overestimation when using observations from the 880 nm wave band (Bond et al., 2013), given the statistically significant decrease in radiative forcing compared with observations at 470nm and substantial decrease compared with observations at 520nm and 660nm. Exceptions are observed when there is a substantially high number loading in the smaller sizes, with forcing values near the UV range yielding a higher radiative forcing. This finding is consistent with new approaches in the literature which use UV observations from TROPOMI

and OMI to improve the overall fit of aerosols (Tiwari et al., 2025; Liu et al., 2024a) to further support that using multiple waveband observations in tandem to disentangle non-linearities in AA's impact on radiative forcing.

Similarly, the radiative forcing low bias computed using a three-modal distribution is smaller than the radiative forcing low bias computed using a more standard log-normal distribution (Wang et al., 2023a), indicating that increasing the resolution of AA size distribution is essential to reducing radiative forcing uncertainty. Biases induced (1.54 to 1.86 W m$^{-2}$) are comparable

to or larger than the uncertainties of observed AOD, SSA, and AAOD used to compute radiative forcing via AERONET or models used by the IPCC to compute aerosol radiative forcing (Schulz et al., 2006; Elsey et al., 2024). Differences caused by applying an intermediate complexity tri-modal size distribution leads to radiative forcing uncertainties in the range from 0.73 to 0.88 W m$^{-2}$, which is similar to impact of methane on radiative forcing over the region (Lu et al., 2025; Hu et al., 2024). For this reason, adapting a trimodal approximation will yield a result which is substantially less good than the actual data, but still

closer to reality than relying on a uniform lognormal size distribution.

A set of rapid regression approximations of the radiative forcing results demonstrate that using observations of SSA, AOD, and particle size in tandem can deliver a fast approximation of the radiative effects herein. Since there are many widely used



and readily available multi-waveband AOD and SSA observations available from satellite, it is hoped that this can lead to a rapid way for existing models to better capture these effects, especially so in regions of the world where pollution is more significant. These findings highlight the need for future models to explicitly account for these factors to better capture aerosol behavior.

One caveat is the length of observations is not long enough to encompass different seasons of the year, although the observations times do capture different meteorological conditions including both inter-basin and intra-basin atmospheric transport. A second caveat is that the observations were made only made at one location within the basin, although the overall range of values observed are representative of industrial regions around the globe with SSA from 0.4-0.78 and AOD from 0.2-1.4 (at 936nm) and from 0.8-2.2 (at 440nm). Future work using additional observations to analyzing the components of the aerosol shell (i.e., nitrate, sulfate, ammonium, water, and secondary organic aerosols) in more detail would aid interpretation and knowledge of how the atmosphere behaves in similar regions, although the current size information of the shell could be used by existing modeling studies to constrain the total sum of these species. Given the substantial differences in constrained SSA and aerosol column number concentration, and resulting increase in radiative forcing, it is hoped that even given the limitations herein, that these results can form the basis of comparison for existing studies, as well as a set of future approaches that other studies can adopt.

### Data and Code availability

The ERA-5 datasets are available at https://cds.climate.copernicus.eu/cdsapp#!/dataset/reanalysis-era5-pressure-levels?tab=overview. Codes is available at https://figshare.com/s/249028a8ae3dc41a7f44. All data are freely available for download at https://figshare.com/s/249028a8ae3dc41a7f44.

### Acknowledgements

We would also like to thank Chien Wang (Laboratoire d'Aérologie, University of Toulouse III – Paul Sabatier, Toulouse, France) and Ralph Kahn (Laboratory for Atmospheric & Space Physics, University of Colorado, Boulder, Colorado, USA) for their assistance reading through the paper and providing many insightful comments and suggestions.

### Author Contributions

**L.Y.G.** was responsible for data curation, formal analysis, software, visualization, and writing the original draft. **J.B.C.** was responsible for conceptualization, funding acquisition, investigation, methodology, project administration, resources, supervision, validation, writing the original draft, reviewing and editing. **S.W.** was responsible for data curation, investigation, and software. **P.T.** and **Z.W.L.** were responsible for investigation, validation and writing the original draft. **K.Q.** was responsible for scientific interpretation and communication, reviewing and editing.

### Competing interests

The authors have the following competing interests: One of the authors is a member of the editorial board of ACP.

### Financial support

This research has been supported by the Fundamental Research Funds for the Central Universities (2024QN11067).

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
