# Peer review of "In-Tandem Multi-Waveband Particulate Absorption and Size Observations Yield Substantial Increase in Radiative Forcing over Industrial Central China"

_EGUsphere, 2025_

## Author Comment (AC1)

Dear Editor and Reviewers,

Thank you for taking the time to provide all of your meaningful and insightful comments and suggestions. We have taken them all into serious consideration and have strived to work hard to address them, in terms of both what was asked as well as in terms of what may further be implied. In this response, your original comments are completely unedited and are given in yellow highlight, our responses are given in blue highlight and updates to the paper are given in pink highlight.

Best Regards,

Jason Cohen

**Reviewer1:**

This study combined optical particle size and multi-band in-situ BC mass and column aerosol optical depth, with MIE modeling to simulate optical properties per particle and over the atmospheric column for absorbing aerosols, providing a realistic probabilistic framework to quantify BC aging and mixing induced optical properties in industrial regions. Valuable results have been shown. In principle, I would suggest its acceptance with necessary revisions.

1. Line 25-26, In addition to relatively old studies, recent studies could also be cited as supporting material, such as Chen et al., (2024, doi: 10.1038/s41467-024-52255-z) for direct climate effect.

Thank you for this useful suggestion. We have added the recommended citation and clarified the climatic context of absorbing aerosols. The following sentence has been inserted in the Introduction:

Aerosols are among the most important forcing agents contributing to anthropogenic global change, producing direct, semi-direct and indirect radiative effects that alter Earth's energy balance (Jacobson, 2001; Bond et al., 2013; Garrett and Zhao, 2006; Chen et al., 2024).

2. Line 31-34, Regarding the solar radiation absorption of BC along with its impact on atmospheric thermal structure has also been examined by a recent study of Pei et al. (2025, doi: 10.5194/acp-25-10443-2025), which is also worthy to mention here.

We thank the reviewer for this valuable suggestion. Previous work has demonstrated that strong absorption by biomass-burning aerosols can generate anomalous heating in the middle troposphere (Pei et al., 2025), rather than being confined to the boundary layer. This vertically heterogeneous heating can substantially modify the thermal stratification and influence large-scale dynamical responses, especially over elevated terrain. We have revised the Introduction to reflect this physical understanding. The following sentence has been inserted in the Introduction:

A subset of aerosols strongly absorbs incoming solar radiation (Ramanathan et al., 2001; Chen et al., 2022a; Pei et al., 2025), and therefore have a further impact on the atmospheric energy balance.

3. Line 35-36, Actually, BC is not only from industrial regions, but also from natural sources such as biomass burning.

We thank the reviewer for highlighting this important point. We agree that black carbon (BC) originates from both anthropogenic and natural sources of combustion (e.g., fossil-fuel use, biomass burning, and wildfires). Our study focuses on industrial regions because our measurements and

analysis target environments with strong co-emissions of semi-volatile species and potential coal-dust interference. We also mention industrial regions, because BC emitted from coal (or other combustion) is clearly anthropogenic in nature, and therefore more directly related to global climate change mitigation, while it is also important to understand natural sources, because these have likely been present before the industrial revolution and hence form a background source of BC. To avoid misunderstanding, we have revised the Introduction accordingly.

BC arises from both anthropogenic and natural sources of combustion (including but not limited to fossil-fuel use, biomass burning, and wildfires).

4. Line 39-41, The sentence could be rephrased to make it easier to follow.

Thank you for the comment. We agree that the sentence could be made clearer. It has been revised in the manuscript as follows:

In industrial regions (such as examined in this study), incomplete combustion and high-temperature processes co-emit diverse carbonaceous particles and semi-volatile species. These carbonaceous particles contain BC and BC like species (including coal dust produced during coal mining, transport and storage) spanning various sizes, although possessing intrinsic set of mie absorption coefficients (Khan et al., 2017). However, the co-emitted semi-volatile species condense upon the BC, leading to transformation of the in-situ particles, turn influencing their microphysical and hence optical properties.

5. Line 41-42, "properties" and "varies" cannot be used together.

Thank you for pointing this out. We corrected the subject–verb agreement by changing "Key microphysical properties varies" to "Key microphysical properties vary" and we checked nearby sentences for similar issues.

6. Line 47-50, Gamma distribution is often used by previous studies.

Thank you for the comment. We agree with the reviewer's point. We revised the phrasing for clarity to:

In parallel, remote sensing studies and chemical transport models commonly represent aerosol size using simplified parameterizations, most often a single lognormal fine mode distribution (Reddington et al., 2013; Shen et al., 2019) or other simplified distributions such as gamma (Hansen and Travis, 1974), which can deviate substantially from observed distributions and introduce systematic biases.

7. Line 72, "quantification" and "estimates" cannot be used together.

Thank you for the suggestion. We removed the redundancy between "quantification" and "estimates." A concise revision is:

Quantification of radiative forcing is widely used to assess BC-radiation interactions and their impacts.

8. Line 90, I wonder if there are many industrial areas in part countries of Global South.

We appreciate the reviewer's constructive comment. Our intention was not to generalize but to highlight that coal-based and heavy industrial regions similar to Shanxi Province are also common in several rapidly developing economies within the Global South, such as India (Brooks, et al.,

2019), Indonesia, South Africa, and Bangladesh where fossil-fuel-based industrial activities are both widespread as well as a major source of absorbing aerosols. To avoid overgeneralization, we have rephrased the sentence in the revised manuscript to make the statement more precise.

To address the aforementioned challenges, this work employs measurements of particle number and multi-band in-situ measurements of BC mass and column AOD over a coal-based industrial region in Shanxi Province, China, which is representative of several rapidly industrializing regions in the Global South, such as those in India, Indonesia, South Africa, and Bangladesh.

9. Line 116, I am curious if it is right that the BC absorption is strong across various wavebands.

Thank you for the reviewer's insightful comment. We agree that the previous wording "strong absorption across various wavebands" could be misleading, as black carbon (BC) exhibits broadband but only weakly wavelength-dependent absorption, rather than uniformly strong absorption at all wavelengths. As demonstrated in previous studies (Bond and Bergstrom, 2006; Sun, 2007; Chen, 2010), BC maintains a nearly constant imaginary refractive index ($k \approx$ 0.7-1.0) from the visible to near-infrared range, indicating relatively consistent absorption across these spectral bands. Here, $k$ physically represents a material's intrinsic light-absorption capacity (the larger the $k$, the stronger the absorption), so a nearly constant k implies weak wavelength dependence. In contrast, brown carbon and dust show strong wavelength dependence, with absorption that decreases markedly toward longer wavelengths (in the case of brown carbon) and that increases towards longer wavelengths (dust). However, in this work when we refer to BC, we are referring to the in-situ mixture of BC which already has come into some equilibrium with its environment, and therefore is coated with a mixture of semi-volatile species like sulfur, nitrogen, and organics, as well as water vapor. Under these conditions, the net per-particle absorption is still broadband, but has some non-linear variation across various wavebands. We have revised the sentence accordingly to improve both clarity and physical accuracy. The revised sentence is as follows:

A multiwavelength Aethalometer (AE-31) was utilized to compute BC mass information based on in-situ environmental conditions, where it has broadband absorption but with some non-linear wavelength-dependence in absorption across the spectrum from the ultraviolet (UV) to the near-infrared (Yang et al., 2009; Bond and Bergstrom, 2006), specifically observed at 370, 470, 520, 590, 660, 880, and 950 nm.

10. Line 124, Note that the Beer-Lambert law is an approximate solution to AOD, without considering multiple scattering, which might be worthy to mention?

Thank you for the helpful suggestion. We have added a clarification noting that our AOD retrieval based on the microtops instrument invokes the Beer-Lambert law as a single-scattering approximation that neglects multiple scattering (and surface-atmosphere coupling). However, the SBDART radiative transfer model that we use for the radiative forcing calculations uses the aerosol SSA and ASY as well as surface spectral albedo in tandem to compute upward and downward fluxes, and thus includes the effects of multiple scattering, approximated by four-stream approach. The revised sentence is as follows:

Additionally, a portable solar photometer (Microtops II) is employed to measure column AOD and precipitable water vapor from direct-sun measurements based on the Beer-Lambert law (Ichoku et al., 2002). This approach is a single-scattering approximation that neglects multiple scattering and surface-atmosphere coupling, and it is most reliable under conditions which have a single and thin

aerosol layer, which is mostly true in heavily industrial areas where there is a very large emissions source near the surface, compared with what may advect in at higher elevation.

11. Line 129-131, I wonder if the 2 times standard deviation threshold value is too small. I remember many studies use 3 times standard deviation threshold value. In other words, how do this threshold value affect the analysis results?

We thank the reviewer for this valuable comment. The ±2 standard deviation (2σ) threshold was selected to ensure temporal continuity and to suppress short-term instrumental noise in the high-frequency (5-min) particle size measurements. Since the filtering was applied within a local moving window of five consecutive data points, a 2σ threshold effectively removes transient spikes caused by sensor fluctuations, coincidence errors, or scattering anomalies, while retaining the physical variability of the dataset.

To evaluate the sensitivity of this choice, we reprocessed the dataset using a ±3σ threshold and compared the derived SSA results for the ISSIZE + Nonuniform group. The differences between the two settings were minimal. All percentile differences remain below ~0.01, which is substantially smaller than the SSA variability discussed in the Results section. Moreover, the number of valid data points increased only slightly (from 1579 for 2σ to 1611 for 3σ), and the overall trends and mean values were unaffected. These results demonstrate that the choice between ±2σ and ±3σ thresholds does not materially affect the SSA statistics or the conclusions of this study.

Table 1. Sensitivity of SSA percentiles to the choice of ±2σ and ±3σ filtering thresholds

| | 10th | 20th | 30th | 40th | 50th | 60th | 70th | 80th | 90th |
|---|---|---|---|---|---|---|---|---|---|
| 2σ | 0.4070 | 0.4324 | 0.4422 | 0.4527 | 0.4701 | 0.4925 | 0.5266 | 0.5826 | 0.6639 |
| 3σ | 0.3959 | 0.4298 | 0.4414 | 0.4503 | 0.4672 | 0.4867 | 0.5176 | 0.5697 | 0.6551 |
| Diff | 0.0111 | 0.0025 | 0.0008 | 0.0023 | 0.0029 | 0.0058 | 0.0090 | 0.0129 | 0.0088 |

12. Line 137-139, please rephrase the sentence. It is currently weird with "in order to" at the end.

Thank you for the suggestion. We have revised this issue and improved the linguistic precision. The revised sentence is as follows:

To represent the mixing state of AA in the industrial region, this work adopted two contrasting assumptions for allocating the observed BC mass, intended to bracket the realistic range of atmospheric mixing states.

13. Line 139-140, To me, this sentence is also weird with grammar issue.

Thank you for the suggestion. We revised the sentence at Lines 139-140 for clarity and grammar. The updated wording is:

Under the uniform assumption, the same BC mass is assigned to each particle size bin, representing an initial mixing state for AA shortly after emission into an air parcel containing contributions from diverse sources which has not yet had time to undergo in-situ atmospheric processing.

14. Line 160, TOA should be put behind the "top of atmosphere"

Thank you for the careful suggestion. We have revised the text so that the first occurrence reads "top of atmosphere (TOA)", and we use the abbreviation TOA consistently thereafter throughout the manuscript.

By integrating these two mixing assumptions and three size distribution schemes, this study establishes a flexible and observation-constrained framework to evaluate how assumptions about particle

morphology influence derived aerosol optical properties including the extinction, scatter, asymmetry, and single scatter albedo (SSA) per particle as well as resulting net top of atmosphere (TOA) radiative forcing.

**15. Line 170-174, How large uncertainties could this assumption introduce?**

We appreciate the reviewer's thoughtful question. We acknowledge that assuming a core-shell structure, with BC and coal dust as absorbing cores and sulfate, nitrate, ammonium, and aerosol water as non-absorbing shells, introduces some uncertainty. Real atmospheric BC exhibits a continuum of mixing states, ranging from fractal aggregates shortly after emission and before in-situ processing occurs, through to compact, coated spherical or spheroidal particles after aging. Sensitivity tests and previous studies (e.g., Jacobson, 2001; Bond et al., 2013; Lack et al., 2009; Tuccella et al., 2020) indicate that the core-shell assumption can lead to variations of about 0.02-0.05 in single scattering albedo (SSA) and 10-30% in top-of-atmosphere radiative forcing, depending on coating thickness and refractive index.

However, it is important to note that this work was performed in a heavily polluted industrial regions, where high concentrations of $SO_2$, $NO_x$, aerosol water, and secondary organic species promote rapid condensation and coagulation, BC particles are expected to be predominantly aged and internally mixed. Under these conditions, the core-shell representation has been shown to reproduce observed absorption and SSA more effectively than externally mixed or purely fractal assumptions. Furthermore, observations of polarization indicate that the vast majority of radiation in this region is polarized, meaning it is interacting with spherical or semi-spherical shapes, which are consistent with the core-shell assumption (Li et al., 2018).

Moreover, in this study the uncertainty associated with particle morphology is partially mitigated by explicitly exploring a wide range of particle sizes, mixing states, and multi-wavelength optical constraints. While the core-shell assumption may bias absolute absorption values to some extent for a fixed size, its impact on the relative differences among size distributions and mixing scenarios examined here is limited, in part because it can reproduce a similar amount of absorption if the size is allowed to vary (Liu et al., 2023; Wang et al., 2019). As a result, the main conclusions regarding the sensitivity of column absorption and TOA radiative forcing to size-resolved BC mass allocation and mixing state remain robust despite the simplified morphological assumption.

**16. Line 187 along with other places, please check reference format to make them suitable.**

Thank you for pointing this out. The inconsistencies were caused by an EndNote export/style issue. We have now audited and corrected all in-text citations and reference-list entries to conform to the journal's style, including the item at Line 187 and all other affected locations.

**17. Line 202, "sources" and "is" – grammar error.**

Thank you for pointing this out. We corrected the subject–verb agreement by changing "sources is" to "sources are", and we reviewed the manuscript for similar issues to ensure consistency.

**18. Line 207, TOA has already been defined earlier in this study.**

Thank you for the note. We removed the redundant re-definition at Line 207. And we have checked the rest of the text to ensure consistent usage.

**19. Line 211-213, Description writing issues! It seems that there are still many writing issues, with**

some of them pointed out here and earlier. I would suggest that the authors make a careful writing revision.

Thank you for this overarching comment. We corrected grammar and removed redundancy, improved subject-verb agreement and parallel structure, and clarified the logical flow with tighter transitions. These revisions improve readability without altering the scientific content.

Net TOA radiative fluxes were simulated separately for the two aerosol-mixing assumptions, using BC amounts and properties inferred from multi-spectral observations. We then analyzed the radiative forcing across these scenarios to evaluate how aerosol mixing state and spectral observation configurations influence the atmospheric energy balance.

20. Line 215-219, Why do the authors select these three statistics?

We thank the reviewer for the constructive comments regarding the statistical analysis. We agree that a clearer connection between each statistical metric and the main scientific questions, would improve clarity and accessibility.

(1) **RMSE** is used at two distinct but complementary levels to quantify uncertainty in TOA radiative forcing estimates.

First, RMSE is calculated between the ISSIZE-based reference forcing (from observations) and forcing derived using simplified size-distribution assumptions ($Log_1$ and $Log_{123}$), as shown in Table 1. This application of RMSE quantifies the absolute deviation in radiative forcing introduced solely by particle size simplification under otherwise identical optical and radiative transfer settings. The substantially larger RMSE values for $Log_1$ compared to $Log_{123}$ demonstrate that using a single-lognormal representation for the aerosol size distribution in the fine mode introduces a larger forcing difference, while using a set of three lognormal to fit the size distribution in the fine mode significantly reduced forcing difference.

Second, RMSE is used to evaluate the performance of regression-based approaches by comparing radiative forcing directly computed using the Mie-SBDART framework with forcing predicted from different regression formulations. In this context, RMSE measures the prediction error associated with statistical approximation of radiative forcing, rather than uncertainty arising from aerosol microphysical assumptions. Together, these two applications of RMSE provide a quantitative link between aerosol representation choices, statistical modeling strategies, and their combined impact on TOA radiative forcing uncertainty.

(2) **The coefficient of determination ($R^2$)** is used to evaluate the consistency of variability in TOA radiative forcing with physically meaningful drivers, particularly changes in particle number concentration, size distribution, and wavelength-dependent absorption. In the manuscript, R2 is reported when comparing forcing variations across wavelengths and size assumptions. Higher R2 values indicate that forcing variability is systematically explained by aerosol microphysical properties rather than by random variability or compensating errors. These results support the conclusion that differences in TOA radiative forcing across wavelengths are physically driven by size and number effects, rather than being artifacts of observational or model noise.

(3) **Two-tailed pointwise t-tests** are applied in the wavelength-dependent forcing analysis to assess whether differences in TOA radiative forcing between wavelength pairs are statistically significant. As described in the manuscript, statistically significant differences are found between $\lambda = 470$ nm and longer wavelengths (520, 660, 880, and 950 nm), with p-values below commonly used significance thresholds. These tests demonstrate that the observed differences in forcing are not

random fluctuations, but represent robust, systematic differences associated with the choice of wavelength.

Taken together, these three metrics provide complementary constraints on TOA radiative forcing uncertainty. RMSE quantifies the magnitude of forcing-related deviations, $R^2$ assesses the physical consistency of forcing variability with aerosol microphysical properties, and the two-tailed t-test establishes the statistical robustness of forcing differences across wavelengths and size assumptions.

21. Figure 2, I would change "orange" color to others so that it could be clearer.

Thank you for the helpful suggestion. We agree that the original orange reduced visual clarity. The updated Figure 2 is as follows:

[Figure]

Figure 2: Variability in the size distribution across all 14 bins at different times, categorized based on the ratio between the first and second peaks in the tri-modal distribution. (a) maximum ratio between the first and second peak; (b) 25[th] percentile ratio between the first and second peak; (c) 50[th] percentile ratio between the first and second peak; (d) 75[th] percentile ratio between the first and second peak.

22. Line 295-297, I understand the logic and agree the assumption, but still wonder how would this assumption affect the results?

We appreciate the reviewer's insightful comment. In brief, the higher RF under the non-uniform assumption arises from how BC mass is redistributed across the observed size distribution, which changes (i) the size-resolved BC mass fraction, (ii) the efficiency of absorption and scattering per unit BC mass, and (iii) the resulting column absorption (AAOD) and SSA that control radiative forcing.

(1) Although total BC mass is conserved between the two assumptions, the size-resolved allocation differs. Under the uniform assumption, the same BC mass is assigned to each size bin, which implies that the BC mass fraction is diluted differently in the size bins that dominate particle number. Under the non-uniform assumption, a constant BC-to-particulate mass ratio is imposed across bins; in our dataset, this leads to more BC mass being assigned to the smaller size bins. Because the GRIMM-constrained size distribution is dominated by accumulation-mode particles, even a moderate redistribution of black carbon mass within the observed fine-mode size bins, while preserving both

the total BC mass and the overall size distribution, shifts the absorption-weighted size distribution toward smaller particles that exhibit higher absorption and differing scattering efficiency per unit BC mass. This is a substantial difference from the mass-based approaches used in the past, which tend to emphasize the much lower number concentrations but overall higher mass loadings found at the larger size bins.

(2) This redistribution matters because single-particle absorption efficiency depends nonlinearly on the size parameter $x=\pi D/\lambda$. In the Mie resonance and transition regime relevant for accumulation-mode particles ($x \approx$ 1-5 for the wavelengths considered here), Mie theory shows that $Qabs$ exhibits strong sensitivity to particle diameter and wavelength (Bohren and Huffman, 1983). As a result, redistributing absorption mass toward smaller size bins within this regime can produce a disproportionately large change in absorption at a fixed wavelength. As particle size increases further and $x$ becomes larger, $Qabs$ becomes progressively less sensitive and approaches oscillatory or quasi-saturated behavior. Therefore, allocating more BC mass to smaller accumulation-mode particles increases the absorption efficiency of the column, particularly in the visible and near-UV bands. This mechanism provides the physical basis for why the non-uniform case produces higher AAOD and lower SSA, which directly translate into stronger positive TOA forcing.

(3) Beyond size effects, the non-uniform allocation is also more consistent with a population in which BC is broadly present as an internal component across size bins (i.e., closer to an internally mixed core-shell representation). Internal mixing and coating formation can enhance absorption via the "lensing effect", whereby a scattering shell focuses incident radiation onto the BC core (Bond and Bergstrom, 2006; Lack et al., 2009). While our framework does not attempt to isolate lensing as a separate measurable term, the key point is that a size distribution with higher BC fraction in the fine-size regime provides a larger population of particles for which coating-related enhancement can operate, reinforcing the absorption increase relative to the uniform allocation.

(4) We also acknowledge that mixing state and size redistribution can modify scattering. However, TOA radiative forcing in our conditions is primarily controlled by the balance between absorption and scattering, and our results show that the increase in absorption dominates over any scattering changes under the non-uniform assumption. This is consistent with the systematic shift toward lower SSA (stronger absorption) and higher AAOD inferred under the non-uniform allocation, which leads to higher positive RF in SBDART. The revised sentence is as follows:

The systematically higher radiative forcing obtained under the internal mixing assumption arises from a coupled effect of how BC mass is distributed across particle sizes and how efficiently those particles absorb radiation. The lensing effect increases the absorption of light by aerosols on a per-particle basis, leading to stronger positive radiative forcing. Crucially, this absorption enhancement does not affect all particle sizes equally. In the Mie scattering regime relevant to these sizes, absorption efficiency remains sensitive to both particle size and composition. As a result, under the non-uniform case, assigning more BC mass to smaller particles within this range enhances column absorption efficiency, contributing to higher AAOD, lower SSA, and stronger positive TOA radiative forcing.

23. Line 323, "can presents"?

Thank you for catching this. We corrected the phrasing by changing "can presents a substantial impact" to "can have a substantial impact" at Line 323, and we reviewed nearby sentences for similar issues.

**Reviewer 2:**

This manuscript presents a novel and valuable integration of in-situ, multi-band, and multi-size aerosol observations with Mie modeling and radiative transfer simulations to improve constraints on BC radiative forcing over industrial Central China. The study framework is scientifically strong and the results are potentially impactful. However, several aspects require clarification and strengthening before the manuscript can be considered for publication. In particular, the authors should: (i) clearly define and justify the mixing-state assumptions used in the modeling, (ii) better situate the findings within the context of existing literature, (iii) discuss the representativeness and limitations of the study region and period, (iv) more explicitly link the theoretical framework to its practical implementation in the radiative transfer model, and (v) improve the manuscript's structure and readability for a broader scientific audience. Additionally, a more transparent treatment of uncertainties and generalizability would further enhance the robustness of the conclusions. Addressing these points will strengthen the clarity, rigor, and interpretability of the manuscript and improve its suitability for publication.

Major comments:

1. Introduction: The introduction provides an overview of regional and global issues related to black carbon (BC) emissions, aerosol optical properties, and radiative forcing in industrial areas. The discussion of uncertainties in BC size, mixing, and optical properties is critical and well-motivated. However, the dense citation style and intermittent sentence structure reduce overall readability. Several sentences combine multiple issues and references, making it challenging to track the key point being made. So, I suggest the authors reorganize to separate conceptual motivations (i.e., why size and mixing matter) from the current limitations of models and observations, summarize the challenges in tabular or bulleted form for clarity before outlining the study aims, and clearly state the main hypothesis or focus questions at the end of the introduction.

We thank the reviewer for the constructive and thoughtful comments on the Introduction. In response, we have substantially revised this section to improve clarity, logical flow, and accessibility for a broad, cross disciplinary audience.

Specifically, we have reorganized the Introduction to clearly separate the conceptual motivations for considering aerosol size distribution and mixing state from the discussion of current limitations in models, remote sensing products, and observational approaches. To reduce sentence density and improve readability, long and compound sentences have been simplified, and references have been redistributed to better support individual key points rather than multiple issues within a single sentence. In addition, we have added a concise summary of the major challenges associated with representing absorbing aerosol size, mixing state, and optical properties, which helps to clarify the motivation for the present study prior to outlining the research approach. Finally, we have revised the concluding paragraph of the Introduction to explicitly state the main hypothesis and focus questions addressed in this work, thereby providing a clearer link between the identified knowledge gaps and the objectives of the study.

2. Methods:
(i) Within the 'Site and Instrumentation', the data collection site is well-chosen to represent typical

industrial settings with BC aerosol heterogeneity. Instrument details (Aethalometer AE-31, GRIMM-180, Microtops II) are described with relevant specifications and data cleaning protocols, which is a strength of the section. Although the temporal and spatial representativeness is asserted, but quantitative evidence is not provided (e.g., discussion of site variability or representativeness of campaign duration). The method for excluding outlier data, which uses standard deviations in five-point windows, may miss sustained measurement drifts or biases—it would be helpful to report how many data points were excluded and statistical justifications used.

We thank the reviewer for raising an important point regarding the temporal and spatial representativeness of the observational dataset. The measurement site is located within a coal-based industrial basin (compromising approximately 2% of global coal production, plus considerable coal-use related industries) characterized by dense industrial activity, residential combustion, traffic emissions, and coal-related industry. This combination of sources is typical of many heavily industrialized regions in northern China and other coal-dependent industrial areas around the global south, providing a representative setting for studying BC aerosol heterogeneity under complex emission conditions. With respect to temporal representativeness, although the campaign duration is limited, the observations encompass a wide range of pollution conditions. During the campaign, BC mass concentrations ranged from approximately 319 to 5798 µg m$^{-3}$ (10th-90th percentile: 1076-3028 µg m$^{-3}$) at 880 nm, AOD ranged from 0.8 to 2.2 at 440 nm, capturing both relatively low- and high-pollution conditions. Importantly, the objective of this study is not to derive climatological mean radiative forcing, but to investigate how aerosol size distribution, mixing state, and wavelength-dependent absorption influence per-particle optical properties and inferred radiative forcing. Given the wide range of values observed, this will allow a diverse set of conditions to be analyzed which can support climate models not only in this important region, but in other energy-intensive regions as well as industrial regions throughout the global south. From this perspective, capturing a diverse set of microphysical and optical conditions as well as their driving factors is more critical than long-term averaging. We have revised the manuscript to clarify this scope and to more explicitly discuss the representativeness and limitations of the study region and observation period.

Although the observation period is limited in duration, the dataset captures substantial variability in BC mass concentration and aerosol loading, encompassing both relatively low- and high-pollution conditions that are characteristic of coal-dependent industrial regions.

Thank you for the reviewer's thoughtful comment regarding the outlier removal procedure. After synchronizing all instruments and removing physically unrealistic values such as -999, a total of 1626 valid time steps remained. The ±2 standard deviation threshold was applied within a local moving window of five consecutive measurements in order to suppress short-term instrumental spikes while maintaining temporal continuity in this high-frequency dataset. Using this criterion, 1579-time steps were retained. To evaluate the sensitivity of this threshold choice, we repeated the filtering with a ±3 standard deviation criterion, which preserved 1611-time steps. Despite this small difference in the number of retained values (approximately 2% of the dataset), the resulting SSA statistics were nearly unchanged. For example, in the ISSIZE + Nonuniform case, the 10th-90th percentile SSA values differed by only 0.01-0.02 between the two thresholds, and the temporal patterns and mean values were essentially identical. These comparisons demonstrate that the conclusions of the study are robust to the specific threshold selected, and that using the 2-standarddeviation filter effectively removes transient anomalies without altering the underlying physical variability of the observations and explainability of the results.

(ii) Coming to the aspect of 'Mixing and Size Parameterization', the adoption of both uniform and non-uniform BC mixing state assumptions is justified and provides a valuable bracket around real-world conditions. However, the manuscript could better clarify why these two extremes are sufficient to span possible states and whether any empirical evidence supports favoring one over the other in this region. I suggest the authors to include a schematic summarizing the two mixing-state approaches for non-expert readers. Also, discuss the implications of possible intermediate mixing states, and whether the assumption of complete internal/external mixing might oversimplify actual atmospheric processes.

(1) The uniform and non-uniform assumptions were intentionally selected because they represent the two physically meaningful endmembers of BC possible mixing states. The uniform assumption corresponds to a weakly mixed, low aging state in which BC mass shows no structural dependence on particle size, an appropriate lower bound for freshly emitted or minimally processed BC, in which its mass distribution is related to its emitted mass distribution, which we assume to be uniform due to no other a prior information. In contrast, the non-uniform assumption enforces a constant BC mass fraction across all size bins, which is only achievable when aerosols have undergone substantial condensation from in-situ nitrate, sulfate, SOA, and water vapor as well as coagulation, forming a stable core-shell morphology. This represents the upper bound of a strongly internally mixed, highly aged aerosol population. These two bounding cases naturally bracket the realistic space of mixing states relevant for radiative forcing calculations, with other possible states found somewhere in the middle.

(2) In addition, several observational characteristics of the study region provide direct empirical support that BC is frequently subjected to rapid atmospheric processing. First, the area hosts a dense cluster of coal-fired power plants, coking facilities, steel production, chemical industries, and widespread coal mining activities, all within a basin-shaped terrain (Fig 1.). This combination leads to high emissions of $SO_2$, $NO_x$, $NH_3$, and abundant water vapor, which strongly promote rapid condensation and coating formation on BC particles. Second, the enclosed basin topography significantly enhances particle residence time, allowing freshly emitted BC to undergo efficient condensation, coagulation, and hygroscopic growth. Third, the observed tri-modal size distribution and the pronounced secondary aerosol signature further indicate that extensive atmospheric processing is occurring over multiple different types of sources with different size characteristics, consistently pushing the aerosol population toward internally mixed structures. For these reasons, the non-uniform assumption is more representative of the dominant state during much of the campaign period. Nevertheless, the uniform assumption remains essential as a conservative lower bound, since fresh or weakly aged BC does intermittently occur and therefore may also contribute to the overall uncertainty range in radiative forcing.

(3) We therefore do not interpret either assumption as an exact representation of the true microphysical state. Instead, the non-uniform assumption is used to represent a physically plausible upper bound for aged, internally mixed BC, while the uniform assumption provides a conservative lower bound that remains relevant when freshly emitted or weakly aged BC is present. In reality, the real atmospheric state of BC spans a continuum of intermediate mixing states and that neither

complete internal nor complete external mixing fully captures the morphological diversity of ambient aerosols, although we believe that the vast majority by total number fraction will be close to the non-uniform assumption. The dual-assumption framework adopted here is thus intended to bound the range of possible optical behavior and radiative forcing responses, rather than to prescribe a single "correct" mixing state, consistent with strategies employed in previous BC optical and radiative forcing studies.

We have added text clarifying that intermediate states may exhibit optical properties lying between the two modeled scenarios. A schematic illustrating the conceptual difference between the uniform and non-uniform assumptions has been added to assist non-specialist readers. The schematic contrasts two ways of allocating BC mass across the same observed particle size distribution. These two assumptions are not meant to represent the true atmospheric mixing state, but instead define a reasonable range of boundaries by which to constrain the resulting optical properties and subsequent radiative forcing variation.

[Figure]

**Figure S3. Schematic illustration of the uniform and non-uniform BC mixing-state assumptions used in this study, showing different allocations of BC mass across particle size bins under identical total BC mass.**

(iii) Mie Model and Radiative Transfer: The use of the Mie model is standard for BC optical properties. The selection of refractive indices and the matching of AOD and BC wavelengths is clear. However, the use of the Santa Barbara DISORT model (SBDART) is only briefly described, and the role of ERA-5 data for ancillary parameters is not thoroughly justified. Please provide references or validation for SBDART in similar BC-rich, high-pollution regions, and explain the implications of using spherical particle assumptions for fresh/aged BC, and discuss any associated uncertainties.

We thank the reviewer for the constructive and important suggestions. Below we clarify the role of ERA-5 in SBDART, provide validation evidence for using SBDART in BC-rich and highly polluted regions, and discuss the uncertainty associated with assuming spherical BC particles.

(1) The radiative effect of BC depends not only on optical properties (SSA, ASY, AOD) which we provide from our observations and modeling, but also on the thermodynamic structure of the atmosphere and concentrations of other radiatively important gasses. Using ERA-5 rather than standard climatological atmospheres reduces bias and ensures that radiative transfer calculations reflect the actual meteorological context during the measurement period. We specifically use ERA-5 values of vertically resolved profiles of temperature, pressure, water vapor, and ozone to construct the layered background atmosphere through which radiation propagates (Wang et al., 2025), and are required to run SBDART. This is selected so that we can use a physically consistent,

observation-constrained source of atmospheric state parameters. It is possible that the value of ozone may not be a perfect match given the large amount of emissions of NOx and CO over this region (Li et al., 2023; 2025), which may lead to a small amount of error in the radiative transfer results (Ricchiazzi et al., 1998; Lamy et al., 2018) however we have no better dataset to use.

(2) The Mie-SBDART framework has been shown to reproduce measured radiative fluxes, SSA/AAOD behavior, and BC radiative forcing in regions characterized by intense pollution, strong BC absorption, and complex mixing states. This includes large urban-industrial basins in China and South Asia (Tiwari, et al., 2023; 2025), where pollution levels and BC loadings are comparable to those in our study region. Such applications demonstrate that SBDART performs reliably in BC-rich environments when aerosol optical properties are provided by observations and Mie calculations across a range of optical wavebands and mixing conditions.

(3) We agree that the assumption of spherical particles represents a simplification, particularly for freshly emitted BC, which often exhibits fractal aggregate structures. Such fresh BC typically produces weaker absorption enhancement and different scattering characteristics compared to compact or coated particles. As BC undergoes atmospheric aging through coagulation and condensation of secondary inorganic species, organics, and aerosol water, particle morphology evolves toward more compact, quasi-spherical core–shell structures. Given the very large amounts of co-emitted species within our domain of interest (Kang, Wang, etc. 2025 ERL), we believe that a very large fraction of the total aerosol number is in fact heavily aged. Furthermore, observations of polarization in this part of China reveal that the vast majority of the aerosols are spherical or semi-spherical in nature (Li et al., 2018), meaning that fractal aggregates do not play a substantial role in the column loading in this region. Even assuming that there is a larger fresh particle amount in the atmosphere than observed by the local polarization observations, our approach uses high resolution observations of particle size and variable mixing states, which currently are not capable of being computed using these advanced fractal aggregate approaches, and hence provides in theory the ability to reproduce the same amounts of enhanced scattering and absorption (Liu et al., 2023; Wang et al., 2019) This uncertainty is partially mitigated in our framework by explicitly exploring a range of particle sizes and mixing states, which together bound the plausible optical responses of BC rather than relying on a single fixed representation (which also has uncertainty associated with it). Previous systematic evaluations of BC mixing state and morphology have shown that spherical core-shell representations often reproduce observed absorption and single-scattering albedo more effectively than externally mixed or purely fractal configurations in environments where BC is heavily aged and coated (Tuccella et al., 2020). While future studies incorporating particle-resolved morphology would further refine these estimates, we do not expect the spherical assumption to alter the main conclusions regarding the relative sensitivity of radiative forcing to particle size and mixing-state assumptions at least within the ranges of values that we have observed herein. Perhaps a slightly different percentile of our total solution space would yield a better fit than a different percentile, but we believe that the actual results should be contained within our uncertainty bounds, since we are refining with additional observations simultaneously.

The industrial basin examined in our study is characterized by many different source types (hence emitted particle sizes), long atmospheric residence times, and abundant secondary inorganic species, conditions that promote compaction and coating growth as well as complex size distributions. It is possible that highly dusty days or days with substantial other absorbing aerosol species would

introduce some additional uncertainty, however during the times we observed, no such conditions were experienced.

(iv) Statistical Analysis: The description of statistical metrics (RMSE, $R^2$, pointwise t-tests) is adequate. However, the direct relevance of each metric to the main questions (especially TOA radiative forcing uncertainty) could be emphasized further. Provide a table or schematic linking each metric to its interpretive role in the conclusions.

We thank the reviewer for the constructive comments regarding the statistical analysis. We agree that a clearer connection between each statistical metric and the main scientific questions, would improve clarity and accessibility.

(1) **RMSE** is used at two distinct but complementary levels to quantify uncertainty in TOA radiative forcing estimates.

First, RMSE is calculated between the ISSIZE-based reference forcing (from observations) and forcing derived using simplified size-distribution assumptions ($Log_1$ and $Log_{123}$), as shown in Table 1. This application of RMSE quantifies the absolute deviation in radiative forcing introduced solely by particle size simplification under otherwise identical optical and radiative transfer settings. The substantially larger RMSE values for $Log_1$ compared to $Log_{123}$ demonstrate that using a single-lognormal representation for the aerosol size distribution in the fine mode introduces a larger forcing difference, while using a set of three lognormal to fit the size distribution in the fine mode significantly reduced forcing difference.

Second, RMSE is used to evaluate the performance of regression-based approaches by comparing radiative forcing directly computed using the Mie-SBDART framework with forcing predicted from different regression formulations. In this context, RMSE measures the prediction error associated with statistical approximation of radiative forcing, rather than uncertainty arising from aerosol microphysical assumptions. Together, these two applications of RMSE provide a quantitative link between aerosol representation choices, statistical modeling strategies, and their combined impact on TOA radiative forcing uncertainty.

(2) **The coefficient of determination ($R^2$)** is used to evaluate the consistency of variability in TOA radiative forcing with physically meaningful drivers, particularly changes in particle number concentration, size distribution, and wavelength-dependent absorption. In the manuscript, $R^2$ is reported when comparing forcing variations across wavelengths and size assumptions. Higher $R^2$ values indicate that forcing variability is systematically explained by aerosol microphysical properties rather than by random variability or compensating errors. These results support the conclusion that differences in TOA radiative forcing across wavelengths are physically driven by size and number effects, rather than being artifacts of observational or model noise.

(3) **Two-tailed pointwise t-tests** are applied in the wavelength-dependent forcing analysis to assess whether differences in TOA radiative forcing between wavelength pairs are statistically significant. As described in the manuscript, statistically significant differences are found between $\lambda = 470$ nm and longer wavelengths (520, 660, 880, and 950 nm), with p-values below commonly used significance thresholds. These tests demonstrate that the observed differences in forcing are not random fluctuations, but represent robust, systematic differences associated with the choice of wavelength.

Taken together, these three metrics provide complementary constraints on TOA radiative forcing uncertainty. RMSE quantifies the magnitude of forcing-related deviations, $R^2$ assesses the physical

consistency of forcing variability with aerosol microphysical properties, and the two-tailed t-test establishes the statistical robustness of forcing differences across wavelengths and size assumptions.

We have included a simple table in the supplemental to summarize these paragraphs.

**Table S1: Summary of statistical metrics used in this study and their interpretive roles in constraining aerosol optical properties and TOA radiative forcing.**

| Statistical Index | Quantified Uncertainty / Information | Physical Interpretation | Manuscript Reference |
|---|---|---|---|
| RMSE | Forcing difference due to simplified size distributions ($Log_1$, $Log_{123}$) relative to ISSIZE reference | Indicates sensitivity of TOA forcing to aerosol size representation | Table 1 |
| RMSE | Prediction error from statistical forcing estimates relative to Mie-SBDART results | Separates statistical modeling error from aerosol microphysical uncertainty | Table 2 |
| $R^2$ | Fraction of forcing variability explained by aerosol number, size, and spectral absorption | High $R^2$ suggests forcing variability is physically driven rather than noise-dominated | Table2; Figure 6 |
| Two-tailed t-test | Statistical significance of wavelength-dependent forcing differences | Confirms that wavelength-dependent forcing differences are systematic | Section 3.3 |

3. Results:

(i) Physical and Optical Properties: The tri-modal size distribution finding is a significant result, offering insight into industrial area aerosol complexity. The connection drawn between combustion sources and specific size modes is plausible. There are some clarity issues, namely, the methods for associating emissions sources to each mode are only briefly mentioned. Is this based solely on size, or also on temporal patterns, source apportionment, or supporting chemical data? Also, the use of the term "SSA bias" is confusing in some parts; clarify whether this is bias relative to true values, model assumptions, or other studies.

We thank the reviewer for highlighting this important point. The present analysis does not aim to unambiguously identify emission sources, but rather to characterize how distinct size modes emerge and persist in a complex industrial environment, and how these modes influence aerosol optical properties and radiative forcing. The linkage between size modes and specific emission or formation processes should therefore be viewed as physically plausible interpretations that warrant further investigation, ideally using complementary measurements such as chemical composition, isotopic tracers, or formal source apportionment techniques. In response, we have revised the manuscript to explicitly frame these associations as qualitative and hypothesis-generating interpretations, rather than definitive source attribution.

To avoid overinterpretation, we have modified the relevant sections to (i) soften causal language, (ii) clearly distinguish observation-based results from interpretive discussion, and (iii) explicitly acknowledge the need for future studies to validate these source-mode relationships.

The first mode (FM) captures details from 0.25 μm to 0.5 μm, the second mode (SM) captures details from 0.5 μm to 0.7 μm, whereas the third mode (TM) captures details from 0.7 μm to 1.6 μm. There are multiple sources of emissions associated with coal mining, production, transportation, and utilization processes, which can plausibly contribute to different portions of the observed size distribution (Fig. S3). Aerosols in FM are plausibly associated with high efficiency combustion process, such as power and steel production, aerosols in the SM are more likely due to lower efficiency combustion associated with boilers, chemicals, and coking, whereas aerosols in the TM are more likely due to coal production, dust, or agricultural residue burning. These associations are intended as qualitative interpretations based on particle size characteristics rather than definitive source attribution.

We thank the reviewer for pointing out the potential ambiguity in the use of the term "SSA bias." In this study, "SSA bias" does not refer to deviation from an unknown true value, nor is it defined relative to a specific external product such as AERONET or OPAC. Instead, it denotes the systematic deviation introduced by simplified assumptions (e.g., single-mode size distributions, single-waveband constraints, or fixed mixing states) relative to the observation-constrained reference solutions based on the full in-situ size distribution (ISSIZE) and multi-waveband measurements. We have clarified this definition in the Results section and revised the wording in several places to explicitly state the reference framework, and changed the word bias to difference although we emphasize that these differences are always negative, thereby avoiding potential confusion. The sentences have been revised in the manuscript as follows:
Although the 1-standard-deviation ranges of SSA start to overlap for 0.65 μm and larger particles, the results still show a consistent per-particle negative SSA difference relative to the ISSIZE reference, with SSA values from the alternative mixing-state and size assumptions remaining systematically lower than ISSIZE at higher wavelengths.

These discrepancies are robust across different wavelengths and all times of day, indicating a consistent positive difference relative to the ISSIZE results, with radiative forcing derived from alternative size distributions remaining higher than the ISSIZE-based results within this wavelength range. Other results are shown in the supplementary materials (Fig. S6) for completeness.

(ii) Radiative Forcing Calculations: The section demonstrates a nuanced examination of size and mixing state choices on column number loading and derived radiative forcing. The sensitivity analysis using different size distributions (ISSIZE, Log123, Log1) is a highlight. There is an apparent contradiction between the statements about minor bias being unavoidable in any parameterization, and the claim that their trimodal framework "substantially" increases accuracy—quantifying the difference and its implications for climate modeling would clarify this issue. Also, the interpretation of SSA values and their physical meaning needs more explanation. I suggest the authors to explicitly compare results to literature or standard models (AERONET, OPAC) in a summary table. Also, discuss limitations in separating lensing effects from actual size distribution impacts.
We thank the reviewer for the thoughtful and detailed comments.
First, regarding the apparent contradiction between the statement that minor bias is unavoidable in any parameterization and the conclusion that the trimodal framework substantially improves

accuracy, we note that the quantitative difference between simplified size representations and the trimodal framework is already explicitly reported in the Conclusions, where we show that radiative forcing is consistently lower than the radiative forcing using a simplified size assumption by 1.54-1.86 W m$^{-2}$, which is reduced to being consistently lower than an intermediate-complexity trimodal representation by 0.73-0.88 W m$^{-2}$. We have revised the text to make this distinction more explicit. Specifically, we now clarify that our results have a radiative forcing which is consistently lower, and that using a reduced-complexity representation as compared to a trimodal framework quantitatively reduces these systematic differences.

For this reason, adapting a trimodal approximation will yield a result which is does not fully reproduce the full observational variability and is consistently too high, but nonetheless provides a quantitatively improved and more physically consistent representation than when a commonly used single lognormal size distribution is applied.

**Table S2. Comparison of aerosol optical representations in OPAC, AERONET, and this study, including size dependence, mixing assumptions, and typical SSA ranges.**

| Approach | Optical property treatment | Size dependence | Mixing assumption | Typical SSA range |
|---|---|---|---|---|
| OPAC | Prescribed optical properties based on idealized aerosol types | Fixed (Lognormal) | External or simplified internal mixing. | 0.85-0.95 |
| AERONET | Retrieved effective SSA from sun-sky radiance inversion | Implicit using two lognormal distributions (one in the coarse mode and one in the fine mode) | Implicit (core shell mixing is applied, using a fixed ratio between core and shell as a single effective mixing state) | 0.85-0.92 |
| This study | Observation-constrained, multi-wavelength inversion | Explicit, size-resolved: based on observations; a simplified three lognormal distribution; and a more simplified single lognormal distribution. | Core-shell mixing with flexible core to shell ratios. | 0.28-0.85 |

There were no AERONET or SONET observations present in the area in which this study was done. However, a comparison using the same base modeling system (with different observations) with AERONET systems have been done using AERONET solely on its own (Tiwari et al., 2023), using AERONET and TROPOMI (Tiwari et al., 2025), using AERONET and OMI (Liu et al., 2024), and even using AERONET and surface observations in tandem (Guan et al., 2025 UNDER REVISION). In all cases, the SSA bands were found to contain the SSA values inverted from AERONET, and the ASY values were found to contain the AERONET ASY values in over 90% of cases.

There is some limitation in terms of separating lensing effects from size distribution impacts, since the size distribution is determined optically, and may inherit some uncertainty from whether its optical size is an actual match with the particle size given that the particle lensing may be different from the range assumed by the optical particle sizer, although such observations are thought to not have such a large uncertainty (Moffet and Prather, 2009).

(iii) The regression-based rapid fitting approach is interesting and potentially useful for global models. However, the generalizability of the results beyond the specific region is not tested or discussed in any detail. So, clearly state the limitations and caveats of these generalized corrections when applied to other industrial or mixed-source regions. Also, consider including an error propagation analysis.

We appreciate the reviewer's insightful comment and agree that the generalizability and limitations of the regression-based rapid fitting approach should be made explicit. In the revised manuscript, we clarify that these microphysical conditions are not universally applicable. The accuracy of the rapid-fitting approach depends critically on both column properties (such as the total column number concentration and mass concentration, AOD and to some extent AAOD) as well as per particle properties (particle size distribution, mixing state, SSA, ASY, and to some extent AAOD). So first and foremost, anywhere else in the world which has a similar range of these values may be able to apply the same fitting representation. It is interesting to note that in this study to range of AOD varies from 0.8 to 2.2 at 440 nm, the range of mixing varies from 0.43 to 1.53, and the range of SSA varies from 0.28 to 0.85. The stability of the underlying size distribution, the dominant mode structure, and the characteristic evolution of shell-core configurations all play a role in determining the SSA, ASY, and to some extent AAOD. In environments where particle aging proceeds differently, for example where coatings develop more slowly, where coarse-mode contributions dominate, or where the particle-size spectrum exhibits stronger temporal variability, the spectral sensitivity of SSA and MAC to mixing state may differ substantially from that observed in our region. Under such circumstances where there are substantial differences, the regression coefficients derived here would lose their physical relevance, and the inferred forcing sensitivities could become non-linear or change sign. For this reason, we clarify that the parameterizations presented in this study are physically grounded but regionally specific; and applying the framework elsewhere may be reasonable of the underlying local optical and microphysical conditions are similar, but would otherwise require recalibration using local optical or microphysical observations.

Our results further provide physically interpretable insights into which optical variables are most influential, allowing the approach to serve as a template for constraining radiative forcing sensitivity in other industrial or mixed-source regions. Fortunately, our region's optical and microphysical properties are closer to those found in many parts of the Global South than similar studies performed over North America or Europe, allowing additional regions of the world to access such correction factors. We also hope that by making the process and data open, that users could reconfigure appropriately over their own regions if they have substantial differences.

4. Discussion and Conclusions: The discussion covers the main findings and implications for climate models and remote sensing products. Limitations of the observational period and site dependence are acknowledged, which lends credibility to the work. The text would greatly benefit from a graphic summary or conceptual diagram showing how the new findings could change radiative forcing estimates in global or regional models. Also, clarify to what extent the proposed framework is ready to be assimilated by existing global models, or what further validation is required.

We thank the reviewer for the helpful suggestions to strengthen the synthesis and model relevance of the manuscript.

Following the reviewer's recommendation, we have added a graphic summary to provide an intuitive, cross-disciplinary overview of how our findings translate into changes in radiative forcing estimates. Rather than presenting a procedural workflow, the figure is designed to contrast simplified aerosol size representations with observation-constrained size distributions, highlighting how different assumptions propagate into aerosol optical behavior and radiative interactions.

[Figure]

**Graphical Abstract: Multi-wavelength, observation-constrained aerosol size representation and its impact on radiative forcing in industrial regions.**

We have also revised the Conclusions to clarify the extent to which the proposed framework is ready to be adopted in existing global modeling systems. Specifically, we now state that the framework may be directly assimilated as a full physical module into current global climate or regional models under matching column and optical conditions, but otherwise cannot be so, and instead is meant to inform and constrain parameterizations that govern absorbing aerosol size representation, mixing-state assumptions, and the resulting optical properties used in radiative calculations. The primary near-term pathway for model uptake is therefore through improved parameter choices and simplified parameterizations (e.g., size-distribution representations and their optical implications) that are consistent with the observational constraints demonstrated here.

General Recommendation:
The technical content and the novel combined observational–modeling approach are strong. However, the overall readability and logical flow of the manuscript are impeded by dense writing, long sentences, and the use of specialized terminology without immediate explanation. To improve accessibility, especially for a cross-disciplinary audience, the authors are encouraged to:
(i) Break up long paragraphs and reorganize dense sections into clearer subsections or bullet points where appropriate.
(ii) Ensure that all figures referenced in the main text are clear, interpretable, and directly support the associated discussion. In several cases, references to supplementary figures interrupt the narrative and may confuse readers; these should be streamlined.
(iii) Provide a transparent and comprehensive discussion of all potential sources of bias or uncertainty, including measurement limitations, modeling assumptions, and representativeness of the study region and period.
Addressing these issues will significantly enhance the clarity, coherence, and scientific impact of the manuscript.

We thank the reviewer for the constructive suggestions aimed at improving readability and accessibility for a cross-disciplinary audience. In response, we have undertaken a comprehensive revision of the manuscript to address each of the points raised.

We have carefully revised the manuscript to reduce sentence length and complexity, particularly in the Introduction and Conclusions. Dense paragraphs have been reorganized to improve logical flow, with clearer separation between conceptual motivation, methodological description, results, and implications. Where appropriate, long compound sentences have been simplified and specialized terminology is now introduced with clearer contextual explanation to enhance accessibility for readers from different disciplinary backgrounds. And we have reviewed all figures referenced in the main text to ensure that they are clearly interpretable and directly support the associated discussion. References to supplementary figures have been streamlined and reduced where they interrupted the narrative flow, with key results now more consistently supported by figures in the main manuscript. In addition, a conceptual graphic summary has been included to synthesize the main findings and their implications for aerosol optical properties and radiative forcing, thereby improving the clarity of the overall presentation.

**Reference**

[1] Bohren, C. F., & Huffman, D. R. (1983). Absorption and scattering of light by small particles. New York: Wiley-Interscience.

[2] Bond, T. C., and Bergstrom, R. W.: Light absorption by carbonaceous particles: An investigative review, Aerosol science and technology, 40, 27-67, 2006.

[3] Bond, T. C., Doherty, S. J., Fahey, D. W., Forster, P. M., Berntsen, T., DeAngelo, B. J., Flanner, M. G., Ghan, S., Kärcher, B., and Koch, D.: Bounding the role of black carbon in the climate system: A scientific assessment, Journal of geophysical research: Atmospheres, 118, 5380-5552, 2013.

[4] Brooks, J., Liu, D., Allan, J. D., Williams, P. I., Haywood, J., Highwood, E. J., Kompalli, S. K., Babu, S. S., Satheesh, S. K., Turner, A. G., and Coe, H.: Black carbon physical and optical properties across northern India during pre-monsoon and monsoon seasons, Atmos. Chem. Phys., 19, 13079–13096, https://doi.org/10.5194/acp-19-13079-2019, 2019.

[5] Chen Y, Bond TC. Light absorption by organic carbon from wood combustion. Atmos Chem Phys. 2010;10(4):1773–87.

[6] Jacobson, M. Z.: Strong radiative heating due to the mixing state of black carbon in atmospheric aerosols, Nature, 409, 695-697, 2001.

[7] Kang, J., Wang, S., Cohen, J., Wang, W., Shi, L., Wang, T., Sun, Q., Feng, J., and Qin, K. (2025). Improving estimation of surface PM2. 5 by including satellite observations of gases, aerosols, and radiation in tandem. Environmental Research Letters, 20(12), 124019.

[8] Lack, D. A., D., C. C., S., C. E., Paola, M., T., A. A., Paul, D., and and Onasch, T. B.: Absorption Enhancement of Coated Absorbing Aerosols: Validation of the Photo-Acoustic Technique for Measuring the Enhancement, Aerosol Science and Technology, 43, 1006-1012, 10.1080/02786820903117932, 2009.

[9] Lamy, K., Portafaix, T., Brogniez, C., Godin-Beekmann, S., Bencherif, H., Morel, B., Pazmino, A., Metzger, J. M., Auriol, F., Deroo, C., Duflot, V., Goloub, P., and Long, C. N.: Ultraviolet radiation modelling from ground-based and satellite measurements on Reunion Island, southern tropics, Atmos. Chem. Phys., 18, 227–246, https://doi.org/10.5194/acp-18-227-2018, 2018.

[10] Li, X., Cohen, J. B., Qin, K., Geng, H., Wu, X., Wu, L., Yang, C., Zhang, R., and Zhang, L.: Remotely sensed and surface measurement- derived mass-conserving inversion of daily NOx emissions and inferred combustion technologies in energy-rich northern China, Atmos. Chem. Phys., 23, 8001-8019, 10.5194/acp-23-8001-2023, 2023.

[11] Li, X., Cohen, J.B., Tiwari, P., Wu, L., Wang, S, He, Q., Yang, H., and Qin, K.: Space-based inversion reveals underestimated carbon monoxide emissions over Shanxi, Commun Earth Environ, 6, 357, https://doi.org/10.1038/s43247-025-02301-5, 2025.

[12] Li, Z., Xu, H., Li, K., Li, D., Xie, Y., Li, L., Zhang, Y., Gu, X., Zhao, W., Tian, Q., Deng, R., Su, X., Huang, B., Qiao, Y., Cui, W., Hu, Y., Gong, C., Wang, Y., Wang, X., Wang, J., Du, W., Pan, Z., Li, Z. and Bu, D. : Comprehensive study of optical, physical, chemical, and radiative properties of total columnar atmospheric aerosols over China: an overview of Sun–Sky Radiometer Observation Network (SONET) measurements, Bull. Amer. Meteor. Soc., 99, 739–755, https://doi.org/10.1175/BAMS-D-17-0133.1, 2018.

[13] Liu, J., Cohen, J. B., Tiwari, P., Liu, Z., Yim, S. H.-L., Gupta, P., and Qin, K.: New top-down estimation of daily mass and number column density of black carbon driven by OMI and AERONET observations, Remote Sensing of Environment, 315, 114436, https://doi.org/10.1016/j.rse.2024.114436, 2024.

[14] Liu, J., Wang, G., Zhu, C., Zhou, D., and Wang, L.: Numerical investigation on retrieval errors of mixing states of fractal black carbon aerosols using single-particle soot photometer based on Mie scattering and

the effects on radiative forcing estimation, Atmos. Meas. Tech., 16, 4961–4974, https://doi.org/10.5194/amt-16-4961-2023, 2023.

[15] Moffet, R. C. and Prather, K. A.: In-situ measurements of the mixing state and optical properties of soot with implications for radiative forcing, Proc. Natl. Acad. Sci. USA, 106, 11872–11877, https://doi.org/10.1073/pnas.0900040106, 2009.

[16] Pei, Q., Zhao, C., Yang, Y., Chen, A., Cong, Z., Wan, X., Zhang, H., and Wu, G.: Wildfires heat the middle troposphere over the Himalayas and Tibetan Plateau during the peak of fire season, Atmos. Chem. Phys., 25, 10443–10456, https://doi.org/10.5194/acp-25-10443-2025, 2025.

[17] Ricchiazzi, P., Yang, S., Gautier, C., and Sowle, D.: SBDART: A research and teaching software tool for plane-parallel radiative transfer in the Earth's atmosphere, Bulletin of the American Meteorological Society, 79, 2101-2114, 1998.

[18] Sun H, Biedermann L, Bond TC. Color of brown carbon: a model for ultraviolet and visible light absorption by organic carbon aero- sol. Geophys Res Lett. 2007; 34 (17).

[19] Tuccella, P., Curci, G., Pitari, G., Lee, S., & Jo, D. S. (2020). Direct radiative effect of absorbing aerosols: Sensitivity to mixing state, brown carbon, and soil dust refractive index and shape. Journal of Geophysical Research: Atmospheres, 125(2), e2019JD030967.

[20] Wang, X., Meng, X., Wang, Y. and Cao, Y.: Simulation of the Optical and Thermal Properties of Multiple Core–Shell Atmospheric Fractal Soot Agglomerates under Visible Solar Radiation. The Journal of Physical Chemistry C, 123, 24225-24233, https://doi.org/10.1021/acs.jpcc.9b04909, 2019.

[21] Wang, Y., Zheng, Z., Sun, Y., Yao, Y., Ma, P. L., Zhang, A., ... & Li, W. (2025). Improved representation of black carbon mixing structures suggests stronger direct radiative heating. One Earth, 8(5).

[22] Tuccella, P., Curci, G., Pitari, G., Lee, S., & Jo, D. S. (2020). Direct radiative effect of absorbing aerosols: Sensitivity to mixing state, brown carbon, and soil dust refractive index and shape. Journal of Geophysical Research: Atmospheres, 125(2), e2019JD030967.

[23] Tiwari, P., Cohen, J. B., Wang, X., Wang, S., and Qin, K.: Radiative forcing bias calculation based on COSMO (Core-Shell Mie model Optimization) and AERONET data, npj Climate and Atmospheric Science, 6, 193, 2023.

[24] Tiwari, P., Cohen, J. B., Lu, L., Wang, S., Li, X., Guan, L., Liu, Z., Li, Z., and Qin, K.: Multi-platform observations and constraints reveal overlooked urban sources of black carbon in Xuzhou and Dhaka, Communications Earth & Environment, 6, 38, 2025.